# AUTOFLY: VISION-LANGUAGE-ACTION MODEL FOR UAV AUTONOMOUS NAVIGATION IN THE WILD

**Xiaolou Sun**[1,2*]**, Wufei Si**[2*]**, Wenhui Ni**[1,2*]**, Yuntian Li**[2*]**, Dongming Wu**[3]**, Fei Xie**[6]**,**
**Runwei Guan**[4†]**, Heyang Xu**[1]**, Henghui Ding**[5†]**, Yuan Wu**[2]**, Yutao Yue**[4]**,**
**Yongming Huang**[1,2†]**, Hui Xiong**[4]

[1]Southeast University  [2]Purple Mountain Labs  [3]MMLab, The Chinese University of Hong Kong
[4]The Hong Kong University of Science and Technology (Guangzhou)
[5]Fudan University    [6]Shanghai Jiao Tong University

## ABSTRACT

Vision-language navigation (VLN) requires intelligent agents to navigate environments by interpreting linguistic instructions alongside visual observations, serving as a cornerstone task in Embodied AI. Current VLN research for unmanned aerial vehicles (UAVs) relies on detailed, pre-specified instructions to guide the UAV along predetermined routes. However, real-world outdoor exploration typically occurs in unknown environments where detailed navigation instructions are unavailable. Instead, only coarse-grained positional or directional guidance can be provided, requiring UAVs to autonomously navigate through continuous planning and obstacle avoidance. To bridge this gap, we propose AutoFly, an end-to-end Vision-Language-Action (VLA) model for autonomous UAV navigation. AutoFly incorporates a pseudo-depth encoder that derives depth-aware features from RGB inputs to enhance spatial reasoning, coupled with a progressive two-stage training strategy that effectively aligns visual, depth, and linguistic representations with action policies. Moreover, existing VLN datasets have fundamental limitations for real-world autonomous navigation, stemming from their heavy reliance on explicit instruction-following over autonomous decision-making and insufficient real-world data. To address these issues, we construct a novel autonomous navigation dataset that shifts the paradigm from instruction-following to autonomous behavior modeling through: (1) trajectory collection emphasizing continuous obstacle avoidance, autonomous planning, and recognition workflows; (2) comprehensive real-world data integration. Experimental results demonstrate that AutoFly achieves a 3.9% higher success rate compared to state-of-the-art VLA baselines, with consistent performance across simulated and real environments. The model, data and code are publicly available at: `https://xiaolousun.github.io/AutoFly`

## 1 INTRODUCTION

Vision-language navigation (VLN) (Chen et al., 2019; Ku et al., 2020; Misra et al., 2018; Krantz et al., 2020; Anderson et al., 2018; Thomason et al., 2020) represents a fundamental capability for Embodied AI, enabling agents to autonomously navigate environments by interpreting natural language instructions alongside visual observations. This capability is particularly crucial for unmanned aerial vehicles (UAVs), which must operate in complex three-dimensional environments while ensuring safe flight dynamics, precise localization, and mission completion under diverse conditions. With UAVs increasingly deployed in diverse applications ranging from search and rescue operations to environmental monitoring and autonomous delivery systems (Aggarwal & Kumar, 2020; Silvagni et al., 2017; Mohsan et al., 2022; Dorling et al., 2016; Chen et al., 2026; Sun et al., 2025; 2024; Wu et al., 2023), the demand for robust navigation systems that can understand and execute human-specified goals has grown substantially.

---

[*]Equal contributions to algorithm design, dataset scenario design, model acceleration, and hardware design
[†]Corresponding author: `huangym@seu.edu.cn`, {`runweig`, `henghui.ding`}`@gmail.com`

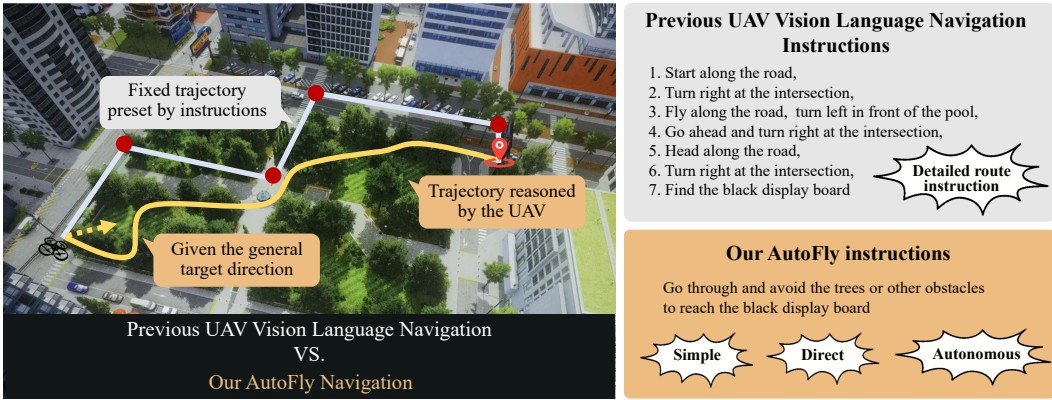

Figure 1: Analysis of previous methods and our AutoFly. *Left*: Previous methods (Lee et al., 2024; Liu et al., 2023b) rely on dedicated, step-by-step instructions that specify predetermined flight paths with explicit waypoints and maneuvers. *Right*: Our AutoFly performs autonomous navigation with concise natural language instructions, and coarse positional or directional information.

Current VLN approaches for UAVs (Fan et al., 2022; Lee et al., 2024; Wang et al., 2024b; Liu et al., 2023b) predominantly rely on dedicated, step-by-step instructions that prescribe predetermined flight paths with explicit waypoints and maneuvers. While these methods have demonstrated effectiveness in controlled environments, they exhibit substantial limitations when deployed in real-world outdoor scenarios with unknown and dynamic conditions. This limitation arises because existing VLN systems typically assume access to comprehensive environmental knowledge and highly detailed navigational instructions, but these assumptions are rarely satisfied in scenarios involving unknown terrain and dynamic conditions (Zhou et al., 2021b; 2022; Xu et al., 2022). In practice, such challenging environments necessitate that human operators provide only coarse-grained directional or location guidance. This discrepancy between research assumptions and real-world constraints creates a critical gap that limits the practical deployment of current VLN systems for UAVs. Moreover, real-world outdoor environments present additional challenges due to their dynamic and unpredictable nature, which require sophisticated autonomous capabilities beyond simple instruction following, as illustrated in Figure 1.

To address these limitations, we introduce AutoFly, an end-to-end Vision-Language-Action (VLA) model specifically designed for UAV autonomous navigation in unknown outdoor environments. AutoFly directly outputs UAV velocity commands to execute high-level action primitives (such as obstacle avoidance, object recognition, and planning) based on coarse guidance. This enables autonomous navigation, powered by a pseudo-depth encoder that derives depth-aware features from RGB inputs to enhance spatial reasoning and multimodal alignment, alongside a progressive two-stage training strategy that aligns visual, depth, and linguistic representations with action policies.

Moreover, existing VLN datasets, while successful for instruction-following applications, present fundamental limitations when applied to real-world autonomous navigation scenarios. These limitations manifest in two critical areas: (1) over-reliance on explicit instruction-following rather than autonomous decision-making, and (2) insufficient real-world data representation, creating a significant sim-to-real gap. To bridge this gap, we present a novel autonomous navigation dataset specifically designed to address each identified limitation. First, we capture trajectories that integrate dynamic planning, obstacle avoidance, and object recognition within continuous workflows, moving beyond the discrete instruction-following paradigms of traditional VLN datasets. Second, we incorporate extensive real-world trajectories to bridge the simulation-reality gap, enabling robust sim-to-real transfer for practical UAV deployment.

In summary, our contributions are threefold: (1) We propose AutoFly, a novel end-to-end vision-language-action (VLA) model for autonomous UAV navigation, featuring a pseudo-depth encoder to enhance spatial understanding from RGB inputs and a progressive two-stage training strategy that seamlessly aligns visual, depth, and linguistic representations with action policies. (2) We construct a large-scale multimodal dataset spanning simulation and real-world outdoor environments with diverse navigation scenarios. (3) Experimental results demonstrate consistent improvements over

state-of-the-art VLA baselines in success rates and maintain consistent performance across both simulated and real outdoor environments.

## 2 RELATED WORK

**Vision-Language Navigation for UAVs**. Vision-Language Navigation (VLN) tasks require agents to follow natural language instructions while interpreting visual cues to traverse environments, a paradigm initially popularized in indoor settings for ground-based robots (Anderson et al., 2018; Vasudevan et al., 2021; Zhu et al., 2020; 2021; Guan et al., 2026b). Extending VLN to UAVs introduces unique challenges due to the three-dimensional dynamics of flight, variable outdoor terrains, dynamic obstacles, and so on. Recent works have adapted VLN frameworks for UAVs (Fan et al., 2022; Lee et al., 2024; Wang et al., 2024b; Liu et al., 2023b), often relying on dedicated, step-by-step linguistic instructions to guide predefined routes in simulated environments like Gazebo (Koenig & Howard, 2004), xView platform (Lam et al., 2018), and AirSim (Shah et al., 2018). These approaches assume access to high-fidelity maps and detailed explicit commands. Recent training-free approaches have reformulated UAV navigation as visual grounding tasks, leveraging the zero-shot capabilities of large-scale pre-trained Vision-Language Models (VLMs) to directly generate 2D waypoints from natural language instructions without requiring task-specific training (Hu et al., 2025; Rajabi & Kosecka, 2025; Qiao et al., 2025). These methods offer compelling advantages in open-world generalization and inherent sim-to-real transferability, but they struggle in dense obstacle environments requiring high-frequency reactive control due to lacking the finetuning.

However, only coarse-grained positional or directional guidance can be provided in real-world unknown scenarios. This gap necessitates models that incorporate autonomous navigation, as addressed in our AutoFly framework, which processes concise instructions and coarse positional information to enable end-to-end navigation in both simulated and real environments.

**High-Level Navigation Primitives for UAVs**. High-level navigation primitives for UAVs encompass foundational capabilities such as path planning, object recognition, and obstacle avoidance, which are essential for safe and efficient flight in complex environments. Traditional approaches typically treat these as separate modular components. For path planning, traditional algorithms like A* or RRT* are adapted to generate collision-free trajectories for UAVs (Karaman & Frazzoli, 2011). More recent approaches in outdoor settings (Zhou et al., 2021a;b; 2022; Xu et al., 2022) integrate path planning with visual SLAM to enable dynamic trajectory generation in unknown environments. Object recognition leverages learning-based algorithms for real-time detection and tracking from onboard cameras (Cao et al., 2021; Sun et al., 2024; 2022). Obstacle avoidance employs reactive methods such as potential fields or learning-based policies trained on depth sensors (Tai et al., 2017; Loquercio et al., 2021; Bhattacharya et al., 2024).

Nevertheless, these primitives are typically implemented as independent modules, resulting in suboptimal performance in unknown environments where dynamic adaptation requires coordinated deliberation across multiple high-level components. To address this limitation, our AutoFly model integrates these primitives, including path planning, object recognition, and obstacle avoidance, into a unified end-to-end VLA architecture that enables joint optimization and coherent decision-making.

**Vision-Language-Action Models**. Vision-language-action (VLA) models (Ahn et al., 2024; Bharadhwaj et al., 2024; Kim et al., 2024; Brohan et al., 2023; 2022; Ding et al., 2024), which integrate visual information and linguistic descriptions to generate executable actions, have gained widespread adoption in robotics, particularly for manipulation tasks. Most current approaches (Ding et al., 2024; Kim et al., 2024; Wang et al., 2024b; Mao et al., 2024) leverage established architectures that combine visual foundation models for environmental perception with textual embeddings, feeding both into pre-trained multimodal large models to produce executable actions. This paradigm has achieved significant success and catalyzed the adoption of VLA architectures in ground robotics (Mao et al., 2024; Ding et al., 2024; Serpiva et al., 2025), where similar end-to-end approaches for VLN tasks have demonstrated comparable effectiveness.

While VLA models achieve remarkable success in ground robotics, existing approaches typically rely solely on RGB inputs, limiting their spatial reasoning capabilities. To address this limitation, AutoFly introduces a novel pseudo-depth encoder that extracts spatial representations from monocular RGB inputs, significantly enhancing spatial reasoning without requiring additional depth sensors.

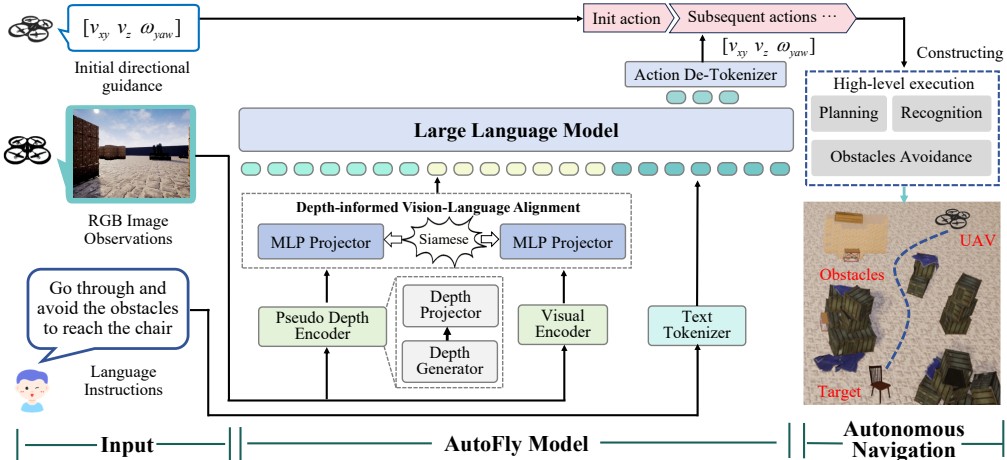

Figure 2: Framework of AutoFly. AutoFly takes RGB observations and linguistic instructions as inputs and directly outputs high-level actions. These actions, combined with initial actions derived from coarse-grained positional or directional information, form action sequences.

## 3 METHOD

This section details our methodology through three key components: VLA model architecture design, autonomous navigation dataset construction, and spatially-informed training paradigm.

### 3.1 TASK FORMULATION

We formulate autonomous navigation as learning a control policy $\pi$ that takes the current RGB observation $o_t \in O$, language instruction $L$, and coarse positional or directional guidance encoded as an initial action $a_0$ as inputs, and outputs a low-level control action $a_t \in A$. The goal is to derive an optimal policy $\pi^* : (O, L) \to A$ that generates a collision-free planning trajectory $\tau$ respecting UAV kinodynamic constraints, ultimately reaching in front of the target, as shown in Figure 1.

### 3.2 VLA MODEL FOR AUTONOMOUS NAVIGATION

UAV navigation poses unique 3D spatial challenges compared to 2D ground robotics. It demands sophisticated geometric understanding beyond RGB-only systems for safe, all-directional obstacle avoidance. UAV applications require enhanced spatial representations due to three key factors: (1) 3D spatial reasoning: UAVs need precise depth estimation for obstacle avoidance, altitude control, and positioning limited in RGB-only systems; (2) Safety-critical navigation: Unlike ground robots, which can tolerate minor collisions, UAV errors risk catastrophic failures, necessitating robust geometric awareness; (3) Scale and distance perception: Accurate distance estimation is essential for safe maneuvers, especially in outdoor settings with ambiguous visual cues.

To enhance geometric reasoning capability, we introduce AutoFly, a VLA architecture augmented with pseudo-depth encoding. Our framework integrates three core components, including a vision-language model, pseudo-depth encoder, and action de-tokenizer, as illustrated in Figure 2.

**Vision-Language Model**. Vision-language models (VLMs) (Liu et al., 2023a; Wang et al., 2024a; Karamcheti et al., 2024; Wu et al., 2025; Guan et al., 2026a) have been extensively adopted in robotics due to their powerful scene-understanding capabilities, language comprehension abilities, and robust generalization properties. In the AutoFly framework, we intentionally employ the LLaVA-based configuration followed by the OpenVLA (Kim et al., 2024; Karamcheti et al., 2024).

**Pseudo-Depth Encoder**. The pseudo-depth encoder generates and encodes depth-aware spatial representations from monocular RGB inputs. It comprises two components: a depth generator that produces depth maps from the RGB inputs, and a depth projector that converts these depth maps into depth tokens aligned with visual tokens from the vision encoder.

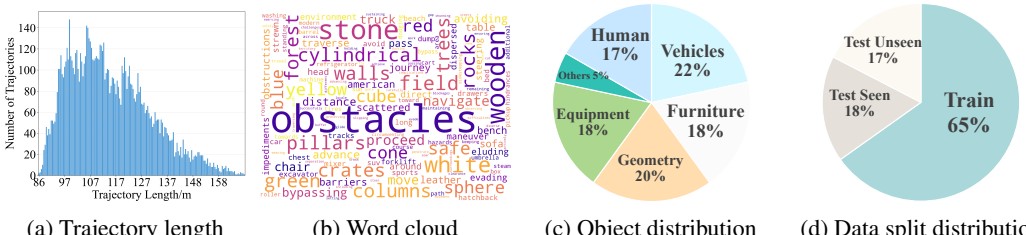

| (a) Trajectory length | (b) Word cloud | (c) Object distribution | (d) Data split distribution |

Figure 3: Overview of autonomous navigation dataset statistical analysis.

Table 1: Comparison of VLN datasets. Datasets for ground robots are shown above the dividing line; aerial-robot datasets are shown below. $N_{traj}$: total number of trajectories; $N_{real\_traj}$: total number of real-world trajectories; $N_{vocab}$: vocabulary size; Path Len: average trajectory length (m); Instr Len: average instruction length; Obs Enc: average number of obstacle encounters.

| Datasets | $N_{traj}$ | $N_{vocab}$ | $N_{real\_traj}$ | Path Len | Instr Len | Obs Enc | Act Space | Sources |
|---|---|---|---|---|---|---|---|---|
| R2R | 7189 | 3.1K | ✗ | 10.0 | 29 | ✗ | graph | ICCV 2018 |
| RxR | 13992 | 7.0K | ✗ | 14.9 | 129 | ✗ | graph | EMNLP 2020 |
| REVERIE | 7000 | 1.6K | ✗ | 10.0 | 18 | ✗ | graph | CVPR 2020 |
| CVDN | 7415 | 4.4K | ✗ | 25.0 | 34 | ✗ | graph | CVPR 2020 |
| TouchDown | 9326 | 5.0K | ✗ | 313.9 | 90 | ✗ | graph | CVPR 2019 |
| VLN-CE | 4475 | 4.3K | ✗ | 11.1 | 19 | ✗ | 2 DoF | ECCV 2020 |
| ANDH | 6269 | 3.3K | ✗ | 144.7 | 89 | ✗ | 3 DoF | / |
| AerialVLN | 8446 | 4.5K | ✗ | 661.8 | 83 | ✗ | 4 DoF | ICCV 2023 |
| CityNav | 32637 | 6.6K | ✗ | 545 | 26 | ✗ | 4 DoF | / |
| OpenUAV | 12149 | 10.8K | ✗ | 255 | 104 | ✗ | 6 DoF | ICLR 2025 |
| OpenFly | 100K | 15.6K | ✗ | 99.1 | 59 | ✗ | 4 DoF | ICLR 2026 |
| **Ours** | **13476** | **147** | **1K** | **107.43** | **12** | **10** | **3 DoF** | **ICLR 2026** |

● *Pseudo-Depth Generator*: Spatial information represented by depth maps plays a crucial role in UAV navigation. The AutoFly framework leverages the robust depth estimation capabilities of Depth Anything V2 (Yang et al., 2024) to generate high-fidelity depth maps. Notably, we deliberately avoid directly using depth maps from depth cameras for two key reasons: first, the depth outputs in AirSim are overly idealized, differing significantly from those captured by real depth cameras, which hinders effective sim-to-real transfer; second, Depth Anything V2 enables the use of only RGB cameras, eliminating the need for specialized depth sensors and thereby reducing payload, cost, and hardware complexity in UAV deployments.

● *Pseudo-Depth Projector*: The depth map produced by the depth generator is first partitioned into patches (Dosovitskiy et al., 2020). These patches are then linearly embedded, yielding the sequence of depth tokens $z_d$. The depth projector serves two primary purposes:

(1) Dimensionality matching: It ensures that the dimensionality of the generated depth tokens ($z_d$) aligns with that of the vision tokens for effective multimodal fusion; (2) Spatial feature alignment: It encodes the inherent spatial information from the depth data and projects it into the visual feature space to achieve better alignment with visual token representations. The spatial encoding enhances the geometric understanding by preserving depth-aware spatial relationships, enabling effective fusion with visual features.

Our pseudo-depth encoder is designed to effectively process depth maps, offering key advantages through the following technical considerations: (1) Domain gap: Depth maps exhibit distinct visual characteristics, such as unique geometric patterns and spatial relationships, which our encoder is tailored to capture accurately. (2) Feature adaptation: Our depth projector explicitly adapts and preserves geometric information from depth data, ensuring seamless alignment with the visual token space. (3) Geometric preservation: Optimized for spatial continuity and relationships critical to navigation tasks, the projector delivers superior depth-aware representations for multimodal alignment.

**Action De-tokenizer**: To enable direct conversion of model outputs into executable actions, we map discrete action tokens to continuous representations. Following OpenVLA's methodology, we utilize the final 256 tokens from the LLaMA2 vocabulary as the action mapping space, as LLaMA2's special token repertoire is insufficient for this purpose. The computation of the action is expressed:

$$\hat{\mathbf{z}}^l = \text{Concat}(\text{Proj}(\text{Proj}_{\mathbf{d}}(\text{DPATv2}(z^{l-1}))), \text{Proj}(\text{SigLIP-DINOv2}(z^{l-1}))), \quad (1)$$

$$\hat{\mathbf{a}}^l = \text{LLM}(\text{Concat}(\hat{\mathbf{z}}^l, \text{Tokenizer}(q^l))), \tag{2}$$

$$\mathbf{a}^l = \text{De-Tokenizer}(\hat{\mathbf{a}}^l), \tag{3}$$

where $z^{l-1} \in \mathbb{R}^{c \times h \times w}$ denotes the image observation of the environment, $\hat{\mathbf{z}}^l \in \mathbb{R}^{m \times c}$ represents the visual embeddings prepared for input to the large language model, $\text{Proj}_{\mathbf{d}}$ denotes the depth projector utilized for depth feature alignment, and $\text{Proj}$ represents the projector employed for vision-language alignment, $\hat{a}^l \in \mathbb{R}^3$ represents the three-dimensional action vector output. $q^l \in \mathbb{S}^k$ denotes a subset of lexical elements from the vocabulary collection $\mathbb{S}$, $a^l \in \mathbb{R}^3$ denotes the three-dimensional action vector generated by the de-tokenizer.

## 3.3 Autonomous Navigation Dataset

**Trajectory Collection**. To generate high-quality trajectories for autonomous navigation tasks, we implement a comprehensive data collection methodology consisting of two interconnected phases: environment construction and trajectory generation.

• *Environment Construction*: We construct 12 diverse simulated environments using AirSim (Shah et al., 2018) for training and evaluation, each spanning 70m×70m and designed to approximate real-world navigation challenges. Each environment incorporates various irregular obstacles commonly found in practical scenarios, including trees, walls, rocks, and buildings, creating complex spatial layouts that demand sophisticated navigation capabilities. For object recognition tasks, we strategically position 60 carefully selected object instances at environment boundaries, with each scenario containing 3-5 distractor objects to challenge the model's recognition and reasoning capabilities. Additional construction details are provided in Appendix A.2.1.

• *Trajectory Generation*: The core objective of autonomous navigation tasks requires the UAV to autonomously navigate from random starting positions at environment edges to the designated target through high-level action primitives, including path planning, obstacle avoidance, and object recognition. While expert human demonstration represents the most effective approach for obtaining optimal trajectories, manual collection proves prohibitively expensive and inefficient for large-scale dataset construction. To address this scalability challenge, we develop specialized data collection agents trained using the Soft Actor-Critic (SAC) RL algorithm (Haarnoja et al., 2018). We train each collection agent in distinct simulated environments until achieving the 95% evaluation success rate. Unlike our AutoFly, these agents are limited to point-to-point navigation without autonomous recognition, language understanding and interaction capabilities. To address this limitation, we strategically position the predefined object instances throughout environments to collect recognition-based navigation trajectories. These automated agents then generate trajectories with performance comparable to expert pilots in terms of navigation success rate. The final dataset combines these automatically generated trajectories with expert demonstration data from corresponding environments, ensuring both scale and quality in the training corpus. In addition to visual observations, language descriptions, and action data, we also collect comprehensive UAV state information, including positional data and attitude angles, to facilitate future research applications. Therefore, we obtain a complete dataset $\mathcal{D}$: $\mathcal{D} = \{\tau_i\}_{i=1}^N$ of trajectories $\tau_i = \{(s_t, a_t, l_i, o_t)\}_{t=0}^{T^i}$, where $s_t \in \mathcal{S}$ is the state (e.g., positional data), $a_t \in \mathcal{A}$ is the action, $l_i \in \mathcal{L}$ is the language instruction, $o_t \in \mathcal{O}$ is the visual observation, and $T^i$ is the length of the i-th trajectory. More details refer to Appendix A.2.3.

**Dataset Rebalancing**. Long-horizon navigation suffers from trajectory imbalance where obstacle avoidance behaviors dominate target-seeking phases, creating learning bias in VLA policies. We quantify this imbalance using KL divergence from the uniform distribution ($\approx 0.36$ nats). To mitigate this issue, we propose a segmentation and rebalancing framework that ensures balanced exposure to diverse navigation behaviors during training. Given a dataset $\mathcal{D} = \{\tau_i\}_{i=1}^N$ of trajectories $\tau_i = \{(s_t, a_t, l_t, o_t)\}_{t=0}^{T^i}$, we segment trajectories using a semantic function $\varphi : \mathcal{L} \times \mathcal{A} \times \mathcal{O} \rightarrow \{1, 2\}$, labeling phases as obstacle avoidance (1) or target seeking (2). Segmentation employs Grounding DINO (Liu et al., 2024) to query visual observations $o_t$ with instructions $l_i$, transitioning phases upon confident detection. For more specifics, see the Appendix A.2.4.

**Dataset Analysis**. Our dataset construction follows systematic design principles to address specific evaluation requirements in UAV autonomous navigation.

• *Dataset Characterization*: Figure 3 (a)-(c) provides comprehensive dataset statistics including trajectory length distributions revealing navigation complexity, word cloud analysis of natural language

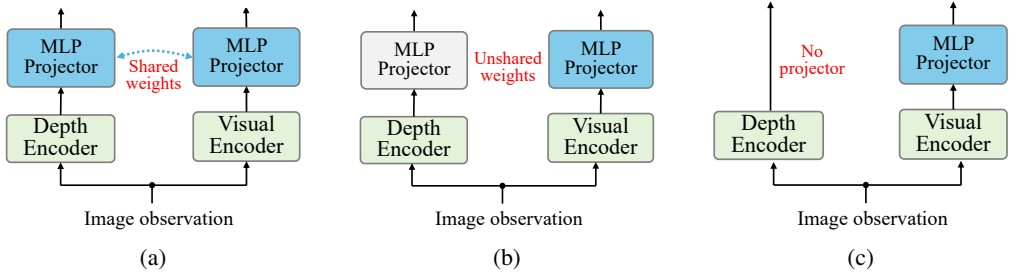

Figure 4: Comparison of three paradigms for integrating depth information during fine-tuning: (a) Siamese MLP projector, (b) Non-Siamese MLP projector, (c) Direct depth integration.

instructions demonstrating linguistic diversity, and target category distributions ensuring balanced representation across object types.

• *Novel Evaluation Metrics*: Unlike existing VLN datasets that primarily focus on navigation success rates, as shown in Table 1, we introduce the average obstacle encounter metric as a quantitative measure of environmental complexity and avoidance capability. This metric captures the challenging nature of our environments and provides objective comparison baselines for obstacle avoidance performance, a critical but under-measured aspect in current VLN evaluation protocols.

• *Simulation-Reality Bridge*: To ensure robust real-world applicability, we collect 1K real-world flight episodes that serve dual purposes: validation of simulation fidelity and augmentation of training data for improved cross-domain generalization. This real-world component addresses the persistent challenge of simulation-to-reality transfer in autonomous navigation systems.

**Dataset Split**. Our training dataset comprises over 13K episodes and 2.5M image-language-action triplets, leveraging both expert demonstrations and simulated trajectories to maximize learning effectiveness (detailed statistics in Table 1 and Figure 3 (d)). For comprehensive evaluation, we conduct experiments across two settings: (1) simulation testing with 7,200 flight episodes covering both seen and unseen scenarios as well as seen and unseen targets, and (2) real-world validation with 200 flights across seen and unseen target conditions. More details refer to Appendix A.2.2.

## 3.4 TRAINING PARADIGM OF AUTOFLY

AutoFly employs a two-stage training paradigm: (1) vision-language alignment, and (2) spatially-informed robot action fine-tuning. We detail each stage below.

**Stage 1: Vision-Language Alignment**. We initialize our model with robust vision-language alignment using the prism-siglip-7b configuration from Prismatic-VLMs (Karamcheti et al., 2024), featuring a two-layer projector and a 7B-parameter LLaMA2 (Touvron et al., 2023). This stage establishes the foundation for multimodal understanding by aligning visual and linguistic representations.

**Stage 2: Spatially-Informed Robot Action Fine-tuning**. Unlike conventional VLA models that rely solely on RGB-language pairs, we introduce a novel training paradigm that incorporates depth information for enhanced spatial reasoning. Our key innovation lies in jointly fine-tuning the pseudo-depth encoder alongside the pre-trained VLA backbone, enabling the model to leverage geometric spatial cues for navigation tasks. We investigate three paradigms for integrating depth information during fine-tuning, as shown in Figure 4: (a) Siamese MLP projector: Maps both depth and visual embeddings into the language space using shared parameters, enforcing consistent cross-modal representations; (b) Non-Siamese MLP projector: Uses separate projectors for depth and visual embeddings; (c) Direct depth integration: Feeds depth embeddings directly to the LLM for implicit alignment. Through empirical evaluation, we find that the Siamese MLP projector achieves superior performance by ensuring consistent feature mapping across modalities, which proves crucial for spatial reasoning tasks requiring precise depth-visual correspondence.

This superior performance arises from the Siamese architecture's parameter sharing mechanism, which enforces unified representational learning across depth and visual modalities. The shared parameters compel both branches to learn consistent feature mappings, naturally preserving spatial correspondence between depth and RGB information while providing implicit regularization to mitigate divergent or conflicting representations.

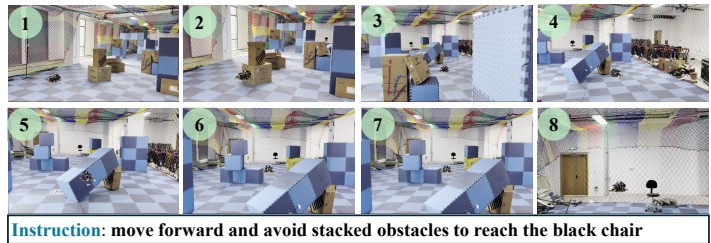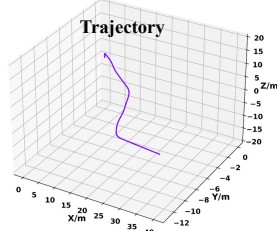

Figure 5: Visualization of AutoFly in the real indoor environment. The experimental arena is a structured indoor environment designed for autonomous navigation and mapping tasks. We have achieved a 60% success rate in real-environment testing. For more visualizations and details, please refer to the Appendix A.6.

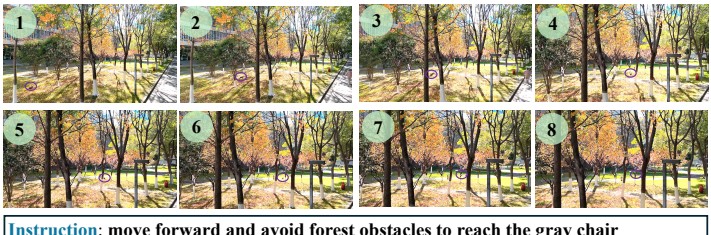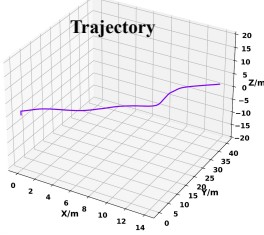

Figure 6: Visualization of AutoFly in the real outdoor environment. The experimental arena is a unstructured outdoor environment with trees. We have achieved a 55% success rate in outdoor forest real-environment testing.

Table 2: Overall performance metrics for quadrotor (all values in %). Here, we report three metrics: Success Rate (SR↑), Collision Rate (CR↓), and Path Efficiency Rate (PER↑). Detailed baseline implementations are provided in the Appendix A.4.1.

| Methods | Seen Scene | | | Unseen Scene | | | Seen Target | | | Unseen Target | | | Overall | | |
|---|---|---|---|---|---|---|---|---|---|---|---|---|---|---|---|
| | SR | CR | PER | SR | CR | PER | SR | CR | PER | SR | CR | PER | SR | CR | PER |
| RT-1 | 26.7 | 61.8 | 62.5 | 18.2 | 79.3 | 59.8 | 25.9 | 61.7 | 61.2 | 26.4 | 57.6 | 60.9 | 24.3 | 65.1 | 61.1 |
| RT-2 | 50.1 | 17.7 | 74.8 | 37.3 | 30.4 | 72.9 | 48.1 | 26.9 | 74.0 | 32.0 | 28.9 | 73.1 | 41.9 | 26.0 | 73.7 |
| OpenVLA | 52.7 | 15.7 | 76.2 | 40.3 | 29.2 | 74.2 | 49.3 | 25.7 | 75.7 | 33.6 | 27.4 | 74.3 | 44.0 | 24.5 | 75.1 |
| **Ours** | **55.4** | **13.0** | **78.1** | **42.3** | **27.2** | **76.5** | **52.3** | **22.7** | **77.9** | **36.4** | **24.6** | **78.1** | **47.9** | **21.9** | **77.3** |

## 4 EXPERIMENTS

### 4.1 IMPLEMENTATION DETAILS

This section presents implementation details across three components: training details, evaluation details, and robot setup. Details for robot setup are provided in the Appendix A.3.1.

**Training Details** The fine-tuning of AutoFly adopts an autoregressive training paradigm where outputs directly correspond to robot action commands. We maintain the default cross-entropy loss function from the base language model and employ a learning rate of 2e-5 for the VLM backbone and 1e-4 for the pseudo-depth projector to fine-tune the model across 80K gradient steps.

**Evaluation Details**. We adopt success rate (SR) as the primary evaluation metric following standard robotics benchmarks. To provide the comprehensive performance assessment, we introduce two additional metrics: collision rate (CR), which measures the proportion of trials where the robot collides with obstacles, and path efficiency rate (PER), which quantifies the average path efficiency across successful trials to assess planning capability. All evaluations follow the dataset split protocol detailed in Section 3.3 and threshold specifications summarized in Appendix A.2.1.

$$\text{SR} = |\mathcal{S}|/N, \quad \text{CR} = |\mathcal{C}|/N, \quad \text{PER} = |\mathcal{E}|/|\mathcal{S}|, \tag{4}$$

where $\mathcal{S} = \{i : d_i \leq d_\tau, \theta_i \leq \theta_\tau\}$ denotes the set of successful trials, $\mathcal{C} = \{i : d_{obs,i} \leq d_{col}\}$ the set of collision trials, and $\mathcal{E} = \{L_{opt,i}/\max(L_i, L_{opt,i}) : i \in \mathcal{S}\}$ the set of efficient trials. Here, $d_i$ is the distance to the target, $\theta_i$ the alignment angle between velocity and target direction, and $L_i$ the

actual trajectory length in episode $i$; thresholds include $d_\tau$ for distance, $\theta_\tau$ for angle, and $d_{col}$ for collisions, with $L_{opt,i}$ as the optimal path length and $d_{obs,i}$ as the distance to the obstacle.

## 4.2 SIMULATION PERFORMANCE

Our experimental results, detailed in Table 2, reveal clear performance patterns across architectures, with RT-1 exhibiting the lowest metrics due to its lack of large language model (LLM) integration for multimodal reasoning. In contrast, LLM-based models including RT-2, OpenVLA, and our AutoFly demonstrate substantial gains in cross-modal understanding for navigation, underscoring the value of LLMs. Both AutoFly

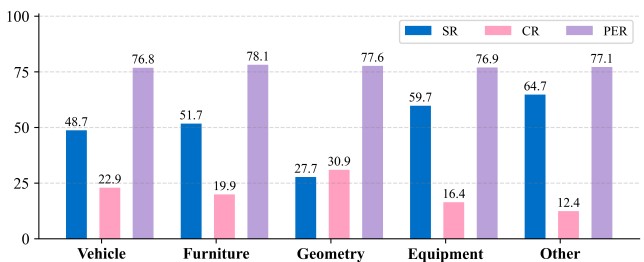

Figure 7: Performance metrics (%) of various distribution of different categories.

and OpenVLA outperform RT-2 across all scenarios, achieving success rates of 47.9% and 44% respectively versus RT-2's 41.9%; this stems from OpenVLA's implicit spatial reasoning via DI-NOv2's visual representations and AutoFly's explicit geometric integration through depth maps. Notably, AutoFly surpasses OpenVLA with a 3.9% higher success rate, 2.6% lower collision rate, and 2.2% improved path efficiency, as shown in Figure 7 for category-specific metrics, highlighting the efficacy of explicit depth cues in dense, obstacle-rich environments.

## 4.3 REAL-WORLD PERFORMANCE

To validate AutoFly's practical deployment capabilities, we conducted comprehensive real-world flight experiments across two distinct environments: a controlled indoor laboratory setting with irregular obstacles and varied object placements, and an unstructured outdoor campus forest environment featuring naturally irregular trees, dynamic swaying branches, and unstructured vegetation. We evaluated

Table 3: Success rate (%) for sim-to-real transferring.

| Scene | Sim : Real | SR | CR | PER |
|---|---|---|---|---|
| indoor | $0K : 1K$ | 10 | 40 | 61.1 |
| indoor | $5K : 1K$ | 25 | 65 | 71.3 |
| indoor | $10K : 1K$ | 60 | 30 | 76.5 |
| outdoor | $10K : 1K$ | 55 | 35 | 75.1 |

the system on 10 object instances, conducting 20 independent trials per target in each setting. As shown in Table refsim-to-real, AutoFly achieves comparable performance across both environments: 60% success rate indoors versus 55% outdoors, with collision rates of 30% and 35%, respectively. The minimal 5% performance gap and similar path efficiency metrics (76.5% vs. 75.1%) demonstrate robust environmental adaptability and consistent navigation capabilities in the presence of dynamic natural elements.

To evaluate transfer learning efficacy, we systematically varied simulation-to-real data ratios. Results show progressive performance gains with increased simulation data, confirming that substantial simulation exposure enhances real-world deployment even with limited real-world fine-tuning. Visualizations of actual flight trajectories are shown in Figure 5 and Figure 6, with deployment details provided in Appendix A.5. For more visualizations, please refer to the Appendix A.6.

## 4.4 ABLATION EXPERIMENTS

We conduct comprehensive ablation studies to validate our model's effectiveness, systematically evaluating five key components: pseudo-depth encoder ablation, specialized depth projector validation, depth-vision-language alignment analysis, dataset rebalancing analysis, and different vision encoder analysis. Analysis of vision encoder variations is detailed in the Appendix A.3.2.

**Pseudo-Depth Encoder Ablation**. To validate the effectiveness of our pseudo-depth encoder, we conduct ablation studies comparing models with and without the depth encoder. The results in Table 4 demonstrate that the method with the pseudo-depth encoder (47.9%, 21.9%) in success rate and collision rate significantly outperforms the one without it (44%, 24.5%), which proves the necessity of introducing spatial information.

Table 4: Results (%) for depth encoder ablation.

| Method | SR | CR | PER |
|---|---|---|---|
| w/ | 47.9 | 21.9 | 77.3 |
| w/o | 44 | 24.5 | 75.1 |

**Specialized Depth Projector Validation**. To validate our design empirically, we compare our specialized pseudo-depth projector against direct application of pre-trained SigLIP to depth maps, using identical protocols and metrics. Table 5 shows our method achieving 47.9% success, 21.9% collision, and 77.3% path efficiency rates, substantially outperforming SigLIP and DINOv2. This gap affirms domain adaptation challenges: while general methods excel on RGB, they fail to extract geometric patterns from depth maps, yielding semantically biased features that hinder spatial reasoning. Conversely, our projector preserves spatial continuity and geometric relationships, delivering superior navigation representations.

Table 5: Results (%) for depth projector ablations.

| Method | SR | CR | PER |
|--------|------|------|------|
| Ours | 47.9 | 21.9 | 77.3 |
| SigLIP | 42.4 | 23.1 | 64.8 |
| DINOv2 | 41.3 | 24.2 | 65.1 |

**Dataset Rebalancing Analysis**. To validate the effectiveness of trajectory rebalancing, we conduct comparative experiments using two configurations, as shown in Table 6: Baseline (no rebalancing) and Rebalanced (with rebalancing). The Baseline setting trains directly on the original imbalanced dataset, while the Rebalanced setting implements our trajectory segmentation and rebalancing framework to address inherent data imbalance in long-horizon navigation trajectories. The Baseline configuration exhibits significantly lower success rates and path efficiency, directly attributable to the severe data imbalance where obstacle avoidance phases dominate training exposure.

Table 6: Results (%) for dataset rebalancing study.

| Method | SR | CR | PER |
|--------|------|------|------|
| w/ | 47.9 | 21.9 | 77.3 |
| w/o | 16.6 | 32.9 | 43.7 |

**Depth-Vision-Language Alignment Analysis**. We conduct an ablation study on depth-vision-language alignment strategies, evaluating three variants in Figure 4: (a) Siamese MLP projector with shared parameters, (b) Non-Siamese MLP projector with separate networks, and (c) direct depth input without projection layers. As shown in Table 7, method (c) yields the lowest success rate (26.7%) due to the modality gap between continuous depth values and discrete LLM tokens, while projection-based methods (a) and (b) achieve markedly higher rates (47.9% and 43.3%, respectively). The superior performance of (a) with a 3.3% edge over (b) validates our hypothesis that parameter sharing enforces consistent cross-modal representations, highlighting the value of unified feature mapping for preserving spatial correspondence.

Table 7: Results (%) for depth-vision-language alignment study.

| Variant | SR | CR | PER |
|---------|------|------|------|
| (a) Siam | 47.9 | 21.9 | 77.3 |
| (b) Non-Siam | 43.3 | 25.3 | 68.2 |
| (c) Direct | 26.7 | 31.9 | 45.9 |

## 5 CONCLUSION

We present AutoFly, an end-to-end Vision-Language-Action model for autonomous UAV navigation using minimal linguistic guidance. Our approach integrates a pseudo-depth encoder for enhanced spatial reasoning, a comprehensive multimodal dataset to support training, and a progressive training paradigm with Siamese MLP projectors for cross-modal alignment. AutoFly achieves 3.9% higher navigation success rates than state-of-the-art baselines, enabling autonomous navigation without detailed instructions. This opens possibilities for real-world applications including search and rescue, environmental monitoring, and autonomous delivery systems.

## 6 ACKNOWLEDGMENTS

This project is supported by the National Natural Science Foundation of China (NSFC) under Grant No. 62225107 and No. 62472104, and by the fund of Hubei Key Laboratory of Inland Shipping Technology under Grant No. NHHY2025001.

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

# A  APPENDIX

## A.1  ABSTRACT

In the appendix, we provide the following supplementary materials:

## A.2  DETAILS OF AUTONOMOUS NAVIGATION DATASET

### A.2.1  DATASET CONSTRUCTION

**Data Collection Framework:** We employ a dual-source approach for dataset construction, combining simulation and real-world data acquisition. Simulation data is collected using 12 custom 70m × 70m scenes constructed in AirSim (Shah et al., 2018), while real-world data is acquired through manual flights in attitude mode within controlled laboratory environments.

**Scene Configuration**: Each simulated scene is populated with diverse obstacles including colored pillars, tree clusters, and stacked boxes, as shown in Figure 8. Object instances from a predefined pool are randomly positioned at scene boundaries, with intra-scene obstacles serving avoidance tasks and boundary targets enabling recognition tasks.

**Evaluation Criteria**: We define navigation success using a dual-criterion approach that ensures both spatial proximity and proper orientation relative to the target. This comprehensive evaluation framework distinguishes between mere proximity-based completion and genuine target-oriented navigation, which is crucial for validating the robot's spatial reasoning capabilities.

**Success Metrics**: A navigation episode is considered successful when the robot achieves: (1) proximity within 5 meters of the target, and (2) target orientation with the angular deviation $\leq 15$ degrees. These thresholds reflect typical UAV operational requirements and ensure the robot demonstrates accurate spatial positioning and proper target alignment for subsequent tasks.

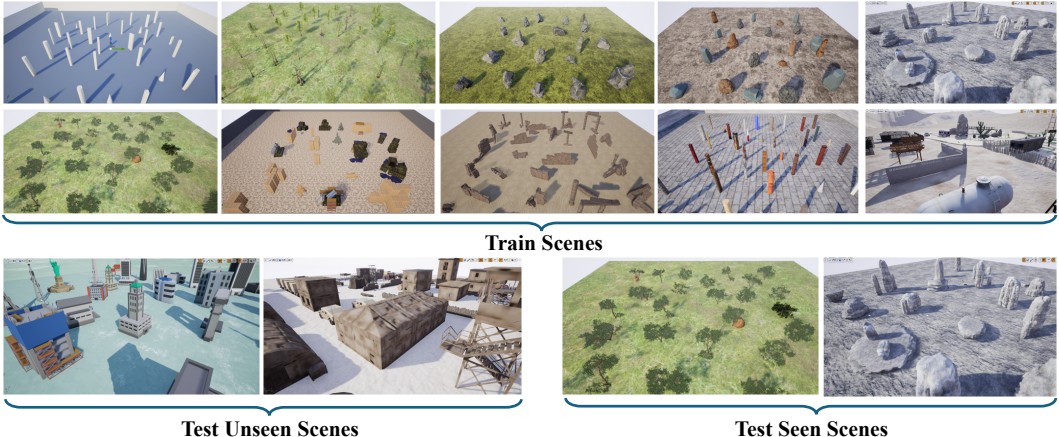

Figure 8: We develop 12 scenarios categorized as 'seen' (10 scenarios) and 'unseen' (2 scenarios), with entirely distinct constituent elements between categories. The 'seen' scenarios provide training data, while evaluation employs both the 2 'unseen' scenarios and 2 reconfigured 'seen' scenarios featuring altered layouts but identical elements, enabling comprehensive assessment of model generalization capabilities.

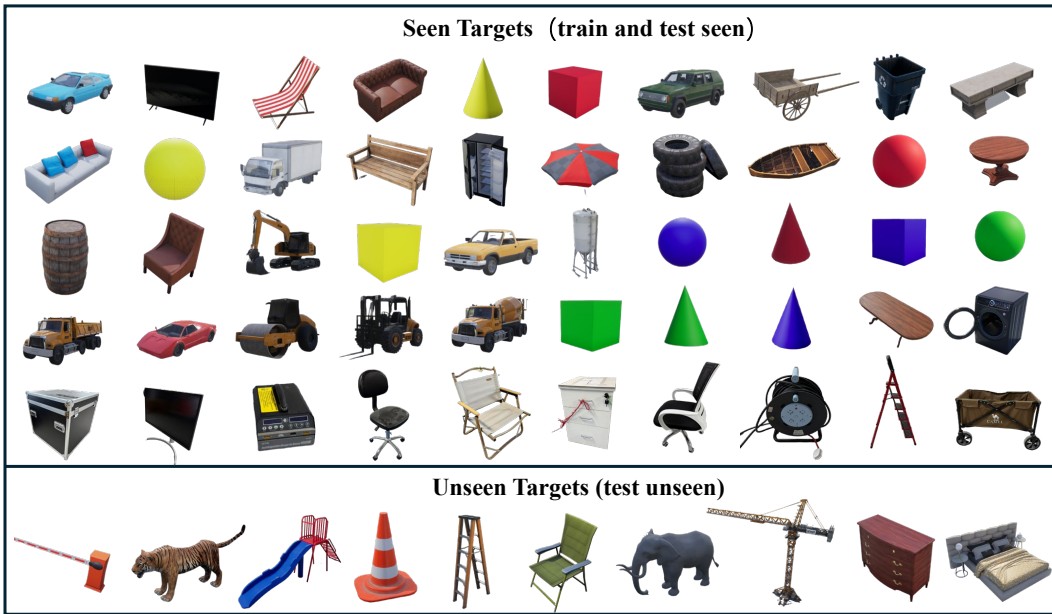

Figure 9: Our object instance design employs a systematic 50/50/10 split across training/test-seen/test-unseen categories, with diverse semantic representation (vehicles, furniture, animals).

### A.2.2 DATASET SPLIT

Our training set comprises 10 scenes with 50 object instances, totaling over 13K episodes and 2.5M image-language-action triplets. Evaluation uses 4 testing scenes (2 from training environments plus 2 completely unseen scenes) with 60 targets (50 previously seen, 10 unseen). This setup creates 4 evaluation conditions, with 30 trials per condition, yielding 7,200 evaluation episodes in total. Each sample follows the format [observation, command, action], enabling comprehensive multimodal learning across planning, obstacle avoidance, and recognition tasks. The experimental scenarios and specific objectives are illustrated in Figures 8 and 9.

### A.2.3 DATA COLLECTION ALGORITHM BASED ON RL

Manual data collection in simulated environments proves prohibitively expensive for large-scale dataset acquisition, necessitating automated alternatives. We initially explore traditional path plan-

ning algorithms (A*, D*) as potential solutions, but find they exhibit significant performance degradation in complex environments with dense obstacles, producing suboptimal trajectories with substantially longer episode durations than expert demonstrations.

To overcome these limitations, we develop specialized obstacle avoidance agents using reinforcement learning, training independent models for each scene. This approach directly outputs velocity commands, eliminating the complexity of trajectory-to-command conversion while maintaining data quality comparable to expert piloting. Our agents employ CNN-based architectures with downsampling layers and MLP heads, trained using Soft Actor-Critic (SAC) with depth-only inputs and stochastic policies. Each trained agent achieves 95% success rate in its respective scene during evaluation, confirming the effectiveness of this automated collection strategy for generating high-quality training data at scale.

The specific supervision process for the critic network is as follows:

$$L_Q(\theta) = \frac{1}{2}\mathbb{E}_{(s,a,r,s')\sim\mathcal{D}}\left[\left(Q_\theta(s,a) - y\right)^2\right],\tag{5}$$

where, $s, a, r, s' \sim \mathcal{D}$ denotes a data sample drawn from the experience replay buffer. $Q_\theta(s,a)$ represents the current Q-network's estimate for the state-action pair $(s,a)$, indicating the expected cumulative reward obtainable after taking action $a$ from state $s$. The term $y$ is the target value used to supervise the Q-network's learning process.

For a stochastic policy, the target value $y$ is calculated as:

$$y = r + \gamma(1-d)\left(\min_{i=1,2} Q_{\theta_i'}(s',a') - \alpha\log\pi_\phi(a'|s')\right),\tag{6}$$

where $\gamma$ is the discount factor and $d$ is a termination flag indicating whether the current state-action pair $(s,a)$ leads to a terminal state. When $d = 1$ (episode terminated), the target Q-value consists solely of the immediate reward $r$. When $d = 0$ (episode continues), the target includes both the immediate reward $r$ and the discounted estimate of future returns. The term $\min_{i=1,2} Q_{\theta_i'}(s',a')$ represents the minimum Q-value estimate from two target networks for the next state-action pair, a technique used to mitigate Q-value overestimation.

The specific supervision process for the actor network is as follows:

$$L_\pi(\phi) = \mathbb{E}_{s\sim\mathcal{D}}\left[\alpha\log\pi_\phi(a|s) - \min_{i=1,2} Q_\theta(s,a)\right],\tag{7}$$

Stochastic policies concurrently optimize both policy entropy and the expected cumulative reward. For the entropy coefficient, the loss is as follows:

$$L_\alpha(\alpha) = -\mathbb{E}_{a\sim\pi_\phi}\left[\alpha\left(\log\pi_\phi(a|s) + \mathcal{H}_{\text{target}}\right)\right],\tag{8}$$

where $\mathcal{H}_{\text{target}}$ represents the target entropy, which is empirically set to $\mathcal{H}_{\text{target}} = -\dim(\mathcal{A})$, where $\mathcal{A}$ denotes the action space and $\dim(\mathcal{A})$ is its dimensionality.

We validate our automated data collection by comparing RL agent trajectories with expert demonstrations in identical scenarios. Statistical analysis across navigation efficiency, safety, and completion metrics confirms comparable quality (Figure 10: a-RL agent, b-expert data), supporting scalable automated collection while maintaining training data quality.

### A.2.4 Dataset Rebalancing

To address the inherent data imbalance observed in long-horizon navigation trajectories, where obstacle avoidance phases dominate over target-seeking phases, we employ a trajectory segmentation and rebalancing framework. This approach ensures balanced exposure to different behavioral modes during training, mitigating biases in policy learning for VLA models.

Given a dataset $\mathcal{D} = \{\tau_i\}_{i=1}^N$ of trajectories $\tau_i = \{(s_t, a_t, l_t, o_t)\}_{t=0}^{T^i}$, we first segment each trajectory into phase-specific sub-trajectories using a semantic segmentation function:

$$\varphi : \mathcal{L} \times \mathcal{A} \times \mathcal{O} \rightarrow \{1,2\},\tag{9}$$

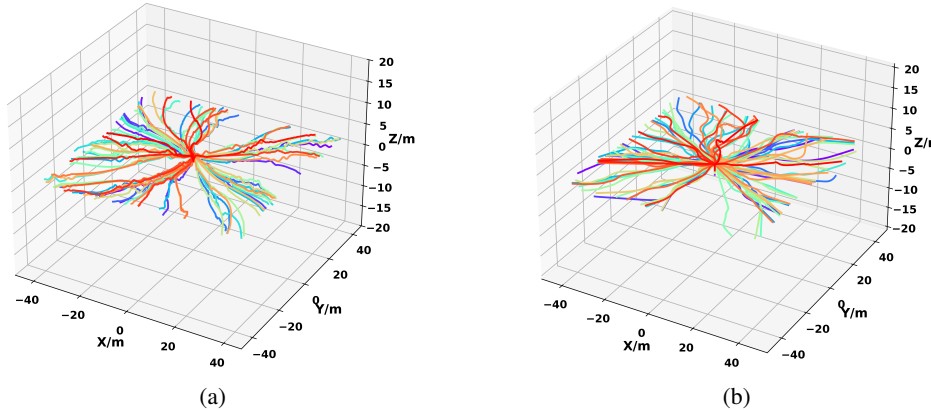

Figure 10: Comparison of expert flight data and data automatically collected by the RL agent in the same scenario: (a) shows the data collected by the RL agent, and (b) shows the expert flight data.

where $\varphi(l_i, a_t, o_t) = 1$ denotes the obstacle avoidance phase, and $\varphi(l_i, a_t, o_t) = 2$ denotes the target-seeking phase.

Segmentation is automated using Grounding DINO (Liu et al., 2024), an open-vocabulary object detector that identifies phase transitions. For each visual observation $o_t$, Grounding DINO uses the language instruction $l_i$ as a query to detect the object. The system transitions from obstacle avoidance phase ($k = 1$) to target seeking phase ($k = 2$) when the target is first detected with confidence exceeding a threshold (e.g., 0.7). Formally, $\varphi(l_i, a_t, o_t) = 2$ if the target is detected, otherwise $\varphi(l_i, a_t, o_t) = 1$.

The original phase distribution is quantified as $P_0(k) = \sum_i |\tau_i^{(k)}| / \sum_i T^i$ and the imbalance is measured via KL divergence from a uniform distribution as follows:

$$D_{KL}(P_0|\text{Uniform}(K)) = \sum_k P_0(k) \log(K \cdot P_0(k)), \tag{10}$$

where in our dataset, this yields $P_0(1) \approx 0.73$ and $P_0(2) \approx 0.27$, resulting in an imbalance of approximately 0.36 nats.

To rebalance, we design a target distribution:

$$P_{\text{target}}(k) = \alpha_k \cdot P_0(k) + (1 - \alpha_k) \cdot \frac{1}{K}, \tag{11}$$

where $\alpha_k \in [0, 1]$ as a balancing parameter (set to 0 for uniform targeting). Resampling weights are computed as $w_k = P_{\text{target}}(k)/P_0(k)$, yielding $w_1 \approx 0.68$ (downsampling obstacle avoidance) and $w_2 \approx 1.85$ (upsampling target seeking) for a uniform target.

We apply stratified resampling as follows: Group sub-trajectories by phase into $\mathcal{D}_k = \{\tau_i^{(k)}\}$, compute sample sizes $n_k = \text{round}(w_k \cdot |\mathcal{D}_k|)$, and sample $n_k$ instances from $\mathcal{D}_k$ (with replacement if $w_k > 1$, without if $w_k < 1$). This produces a balanced dataset $\mathcal{D}'$, theoretically grounded in importance sampling, where expectations under $P_0$ are preserved via weight adjustments:

$$\mathbb{E}_{\tau \sim P_0}[f(\tau)] = \mathbb{E}_{\tau \sim P_{\text{target}}}\left[f(\tau) \cdot \frac{P_0(\tau)}{P_{\text{target}}(\tau)}\right]. \tag{12}$$

## A.3 EXPERIMENTS

### A.3.1 ROBOT SETUP

**Hardware Configuration**. We employ a custom-built quadrotor equipped with a front-mounted RealSense D455 camera, as shown in Figure 11. Although the D455 supports both RGB and depth capture, we utilize only RGB frames at $640 \times 480$ resolution with $90°$ HFOV to validate our pseudo-depth encoder's effectiveness in RGB-only scenarios. The UAV features a distributed computing architecture, with an onboard computer handling real-time odometry and basic flight control, while a Wi-Fi module enables communication with a remote server for intensive VLA model inference.

**System Architecture**. This configuration addresses the computational constraints of embedded systems by offloading VLA processing to remote servers while maintaining real-time localization capabilities onboard. The Wi-Fi communication protocol provides sufficient bandwidth for RGB image transmission and low-latency command reception within controlled indoor environments, ensuring stable operation during evaluation flights.

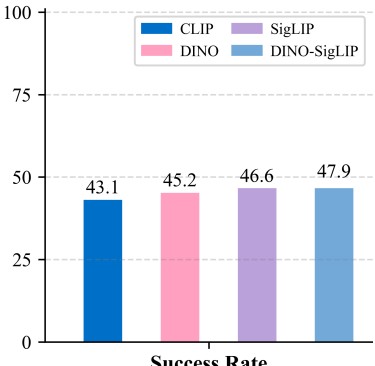

Figure 11: The quadrotor configuration employed in this study features a custom-built platform.

### A.3.2 ABLATION EXPERIMENTS

**Different Vision Encoders Analysis**. To investigate the impact of different vision encoders on task performance, we conduct comprehensive ablation experiments comparing four representative vision backbones: CLIP, DINO, SigLIP, and a fusion approach combining DINO and SigLIP features.

- *Results*: To investigate the impact of different vision encoders on navigation performance, we compare four representative backbones: CLIP, DINO (DINOv2), SigLIP, and DINO-SigLIP fusion, as shown in Figure 12. Results demonstrate clear performance differences: SigLIP achieves the highest success rate among single encoders (46.6%), outperforming CLIP (43.1%) by 3.5% and DINO (45.2%) by 1.4%. The DINO-SigLIP fusion delivers optimal performance at 47.9%, representing a 1.3% improvement over SigLIP alone.

- *Analysis*: The superior performance of DINO-SigLIP fusion stems from the complementary strengths of both architectures. DINO's self-supervised learning approach provides a certain degree of robust spatial understanding essential for 3D navigation tasks, while SigLIP contributes enhanced visual-language alignment that enables more precise interpretation of navigation instructions. This combination effectively leverages DINO's spatial awareness with SigLIP's multimodal fusion capabilities, resulting in more accurate navigation decisions and improved overall performance in complex autonomous navigation scenarios.

Figure 12: Performance metrics (%) for different vision encoder.

### A.3.3 EVALUATION ON CHALLENGING SCENARIOS

While our main experiments demonstrate the effectiveness of the pseudo-depth encoder across diverse navigation tasks. To rigorously evaluate the depth encoder's robustness, we design three representative challenging scenarios that isolate specific navigation difficulties: dense obstacle configurations, irregular natural structures, and dynamic moving obstacles. These experiments aim to quantify the depth encoder's contribution to both safety (collision avoidance) and efficiency (path planning) under stress conditions.

- *Experimental Design*: We construct three distinct, challenging environments: (1) Dense Cylinders Scene: Numerous closely-spaced cylindrical obstacles of varying sizes, creating narrow navigation gaps that demand precise spatial reasoning. (2) Dense Forest Scene: Irregularly distributed obstacles with complex geometries resembling natural forest structures, presenting high visual complexity. (3) Dynamic Obstacle Scenarios: All obstacles are converted to continuously moving objects throughout the evaluation period, requiring predictive spatial reasoning and real-time avoidance. The testing methods and quantities are consistent with those in experiments described in the main text.

- *Results*: Table 8 presents the comparative performance across all three challenging scenarios. The pseudo-depth encoder consistently delivers substantial improvements in both success rates (SR) and collision rates (CR) across diverse environmental conditions, with the most pronounced gains observed in the dynamic obstacle setting.

- *Analysis*: We conduct separate analyses according to scenarios, and we also provide a comparative demo for navigation in different scenarios[1].

---

[1] https://xiaolousun.github.io/AutoFly

Table 8: Overall performance metrics for quadrotor in challenging scenarios (all values in %). Here, we report three metrics: Success Rate (SR↑), Collision Rate (CR↓), and Path Efficiency Rate (PER↑). The last line represents the gain.

| Method | Dense Cylinders Scene | | | Dense Forest Scene | | | Dynamic Obstacle Scenarios | | |
|---|---|---|---|---|---|---|---|---|---|
| | SR | CR | PER | SR | CR | PER | SR | CR | PER |
| w/ | 57.2 | 21.1 | 78.3 | 53.6 | 23.7 | 77.3 | 50.7 | 28.2 | 73.9 |
| w/o | 49.3 | 28.7 | 76.1 | 45.9 | 31.1 | 75.6 | 40.8 | 37.7 | 71.1 |
| Δ | **+7.9** | **-7.6** | **+2.2** | **+7.7** | **-7.4** | **+1.7** | **+9.9** | **-9.5** | **+2.8** |

**Dense Cylinders Scene**: The baseline model exhibits two primary failure patterns: (a) frequent collisions when attempting to navigate through narrow gaps (CR: 28.7%), and (b) overly conservative path planning that leads to excessive detours and step-limit failures. The depth encoder enables accurate distance estimation to obstacles and gaps, allowing the UAV to confidently identify safe passages. This spatial reasoning capability translates to a 7.9 % improvement in success rate and a 7.6 % reduction in collision rate.

**Dense Forest Scene**: In this visually complex environment, the baseline model demonstrates hesitant navigation behavior, struggling to parse irregular obstacle geometries. The resulting high collision rate (31.1%) indicates poor spatial understanding. The depth encoder's geometric reasoning significantly enhances performance, reducing collisions by 7.4 % while improving planning efficiency by 1.7 %. This enables more decisive navigation decisions and superior trajectory optimization through visually ambiguous regions.

**Dynamic Obstacle Scenarios**: This extreme test case reveals the most dramatic performance gap. The baseline model's collision rate reaches 37.7%, frequently failing to maintain safe distances from moving obstacles or predict their trajectories. In contrast, the depth encoder enables predictive spatial reasoning, allowing the model to anticipate obstacle movements and execute proactive avoidance maneuvers. The resulting 9.9 % increase in success rate and 9.5 % reduction in collision rate provide the strongest validation of the module's necessity for dynamic environments.

**Cross-Scenario Insights**: The depth encoder's performance gains scale with scenario difficulty. The most substantial improvements occur in the dynamic obstacle setting (+9.9% SR), where temporal prediction and continuous spatial updating are most critical. This validates our hypothesis that explicit geometric reasoning becomes increasingly valuable as environmental complexity increases.

### A.3.4    ANALYSIS OF SIMPLER ALTERNATIVE APPROACHES

To verify that the performance gain of our pseudo-depth encoder cannot be replicated by simpler modifications, we conduct three controlled experiments using alternative strategies: data scaling, data augmentation, and upgrading the vision encoder.

● *Experimental Setup*: All experiments share the same evaluation protocol and test set as the main paper. We compare the following approaches against our baseline (we use the OpenVLA here) and the full AutoFly model:

- **Data Scaling.** We collect an additional 1,000 episodes (∼350K vision-language-action pairs) to evaluate the effect of increased training data.

- **Data Augmentation.** We apply photometric augmentations including random brightness, contrast, saturation, and hue adjustments. Geometric augmentations are excluded to preserve the spatial alignment of action labels.

- **Stronger Vision Encoder.** We replace the SigLIP visual backbone with SigLIP 2 (fused with DINOv2), a state-of-the-art RGB encoder, to assess whether improved visual features can substitute for explicit depth reasoning.

● *Results*: As shown in Table 9, while adding training data yields the largest improvement among the alternative approaches (+1.2% SR), it remains substantially below the +3.9% SR gain achieved by our pseudo-depth encoder. Data augmentation and a stronger vision encoder provide only marginal improvements of +0.3% and +0.7% SR, respectively. These results confirm that the geometric and spatial reasoning capabilities introduced by the depth encoder are essential for the navigation task and cannot be replicated by scaling data volume or upgrading RGB-only perception.

Table 9: Comparison with simpler alternative approaches (all values in %). Here, we report three metrics: Success Rate (SR↑), Collision Rate (CR↓), and Path Efficiency Rate (PER↑).

| Method | SR | CR | PER |
|---|---|---|---|
| Baseline | 44.0 | 24.5 | 75.1 |
| Data Augmentation | 44.3 (+0.3) | 24.1 | 75.5 |
| Stronger Encoder | 44.7 (+0.7) | 23.1 | 76.6 |
| Additional Training Data | 45.2 (+1.2) | 23.6 | 76.5 |
| **AutoFly** | **47.9 (+3.9)** | **21.9** | **77.3** |

## A.4 METHODOLOGY DETAILS

### A.4.1 BASELINE CONSTRUCTION DETAILS

To ensure fair comparison, all baselines are implemented using identical training datasets and evaluation protocols. Given the constraints of our UAV navigation domain, we adapt baseline architectures while preserving their core methodological contributions:

- **RT-1** (Brohan et al., 2022): We follow the original architecture using EfficientNet-B3 for visual encoding and Universal Sentence Encoder for language processing. Visual and language embeddings are integrated via FiLM layers for cross-modal conditioning. The Token Learner mechanism and Transformer architecture remain unchanged from the original implementation.

- **RT-2** (Brohan et al., 2023): We reproduce RT-2's vision-language-action architecture with its core methodological approach of treating actions as text tokens. To ensure compatibility with our evaluation framework and training infrastructure, we employ the prism-siglip-7b configuration from Prismatic VLMs as the backbone, maintaining RT-2's fine-tuning strategy and action tokenizer scheme.

- **OpenVLA** (Kim et al., 2024): Our implementation preserves OpenVLA's key innovations including DINOv2 visual features and the action chunking mechanism. We utilize the same prism-siglip-7b backbone for consistency across VLM-based baselines, ensuring that performance differences reflect methodological contributions rather than backbone variations.

This standardized backbone approach enables fair comparison of each method's core contributions while maintaining implementation feasibility within our experimental framework.

## A.5 DEPLOYMENT

### A.5.1 DISTRIBUTED SYSTEM ARCHITECTURE

Our deployment strategy implements a distributed computing architecture that separates computational processing from UAV flight operations to optimize both performance and safety. The AutoFly model executes on a remote server infrastructure, enabling access to high-performance computing resources while maintaining real-time control responsiveness for autonomous navigation tasks, as shown in Figure 13.

### A.5.2 NETWORK COMMUNICATION PROTOCOL

The system establishes bidirectional communication between the UAV platform and remote server through a dedicated Local Area Network (LAN) connection. The data transmission protocol operates as follows:

- **Uplink Stream**: The UAV continuously transmits visual sensor data (RGB imagery), current state information, and linguistic instructions through the LAN to the remote processing server

- **Downlink Stream**: The server processes multimodal inputs through the AutoFly model and returns computed velocity commands (forward velocity, yaw angular velocity, vertical velocity) to the UAV flight controller via the same network connection

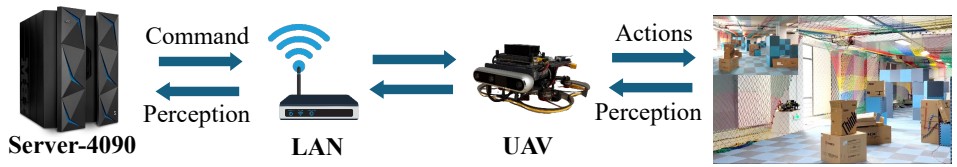

Figure 13: System deployment architecture. Our model is implemented on a remote server and communicates with the robot via a local area network (LAN).

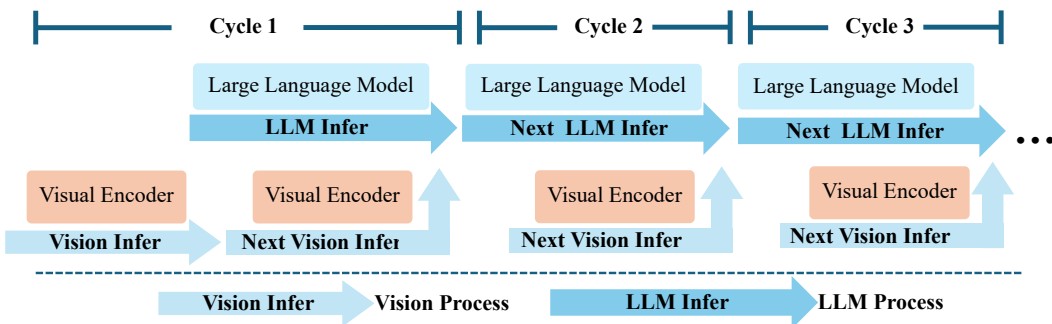

Figure 14: Our system employs a multi-process parallel inference methodology. Two parallel processes: a vision pipeline and an LLM pipeline execute concurrently. After the initial sequential cycle, visual processing for frame $t+1$ overlaps with LLM inference for frame $t$, reducing per-frame latency from 120ms to 85ms and enabling real-time UAV control at 15 FPS.

### A.5.3 MODEL ACCELERATION

To achieve real-time performance for UAV deployment, we implement a comprehensive optimization pipeline for AutoFly. The model is decomposed into modular components (vision encoder, pseudo-depth encoder, depth projector, alignment MLP, and LLM) to enable independent optimization and parallel processing. Each component is converted to ONNX format for cross-platform compatibility and enhanced inference efficiency.

We employ TensorRT for LLM acceleration, which provides significant speedup through kernel fusion and mixed-precision inference. Additional CUDA operators are implemented for custom depth processing operations, while model parallelism enables distributed inference across multiple GPU processes to handle the computational demands of multimodal processing. This optimized pipeline achieves 15 FPS inference on an RTX 4090 GPU, representing a 7.5 × speedup compared to the baseline PyTorch implementation. Control commands are transmitted to the UAV via low-latency network communication, enabling real-time navigation with sub-100ms response times suitable for dynamic flight scenarios.

### A.5.4 PARALLEL INFERENCE ARCHITECTURE

To minimize end-to-end latency, we implement a pipelined multi-process inference system that overlaps visual processing with LLM computation, as shown in Figure 14. The architecture employs two parallel processes: a vision pipeline handling RGB encoding, depth generation, and feature projection, and an LLM pipeline processing multimodal tokens for action prediction. During the initial inference cycle, both visual processing and LLM inference execute sequentially, establishing the baseline latency. For subsequent frames, we exploit the temporal overlap by initiating visual processing for frame $t + 1$ concurrently with LLM inference for frame $t$. This pipelining strategy leverages the observation that LLM inference typically dominates the computational bottleneck in VLA models. The optimized pipeline achieves near-optimal throughput where total inference time approaches the LLM inference duration plus inter-process communication overhead (approximately 15-20ms). This represents a significant improvement over sequential processing, reducing end-to-end latency from 120ms to 85ms per frame, enabling real-time UAV control at 15 FPS.

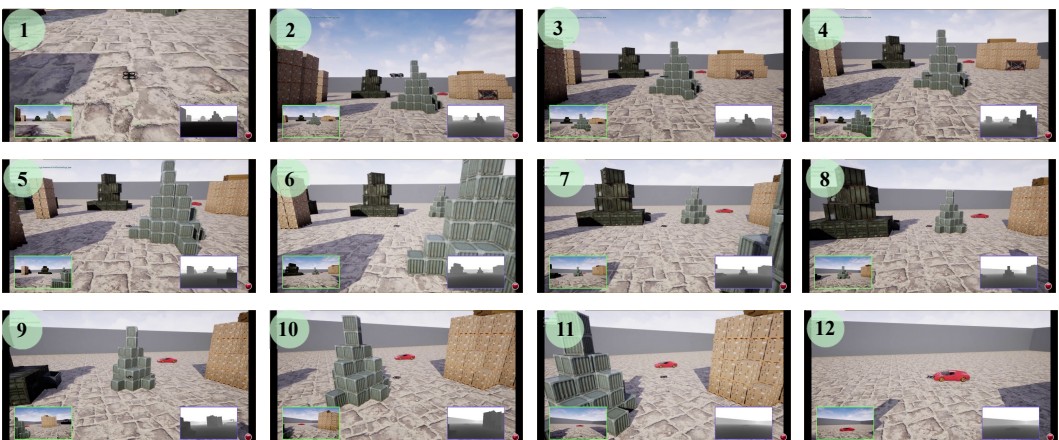

Figure 15: Visualization of AutoFly in a simulated environment. The verbal command is to move forward and avoid stacked obstacles to reach the red sports car.

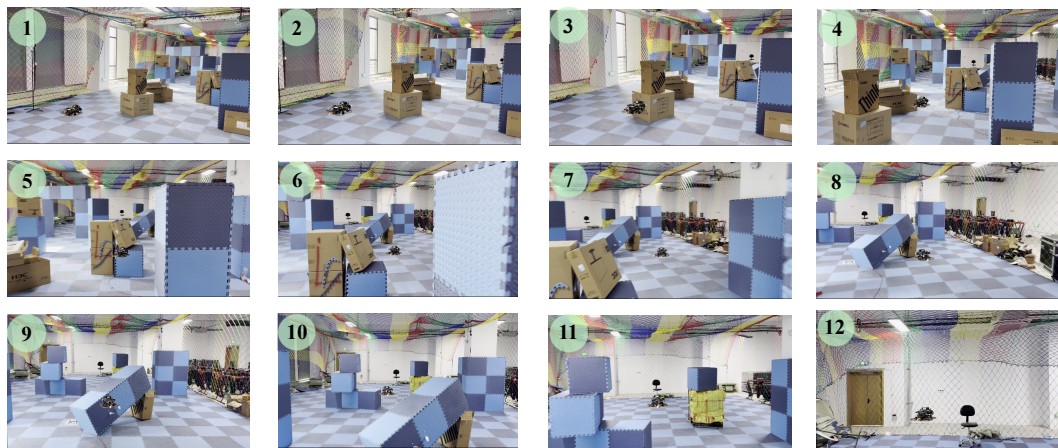

Figure 16: Visualization of AutoFly in a real environment. The verbal command is to move forward and avoid stacked obstacles to reach the black chair.

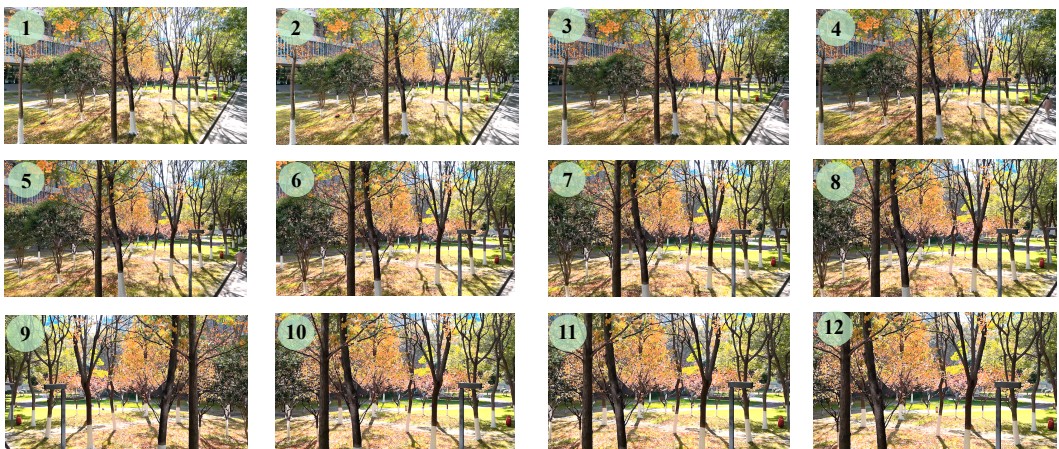

Figure 17: Visualization of AutoFly in a real outdoor environment. The verbal command is to move forward and avoid forest obstacles to reach the gray chair.

## A.6 VISUALIZATION

Figures 15, Figure 16, and Figure 17 present comprehensive visualizations of AutoFly's autonomous navigation performance across both simulated and real-world deployment scenarios. These demonstrations illustrate the model's integrated reasoning capabilities, where the system successfully coordinates multiple navigation competencies within a unified decision-making framework.

### A.6.1 SIMULATION ENVIRONMENT RESULTS

In controlled simulation scenarios, AutoFly demonstrates robust autonomous navigation across diverse environmental challenges. The visualizations reveal the model's ability to simultaneously perform dynamic path planning around complex obstacle configurations, execute real-time collision avoidance maneuvers, and maintain goal-directed behavior toward linguistically specified targets. Notably, the system operates effectively in previously unseen environments, validating the generalization capabilities developed through our comprehensive training methodology.

### A.6.2 REAL-WORLD DEPLOYMENT VALIDATION

Real-world flight demonstrations confirm AutoFly's practical effectiveness beyond simulation environments. The system successfully translates its learned reasoning behaviors to physical UAV platforms, navigating complex outdoor environments while maintaining the integrated planning, perception, and control capabilities observed in simulation. These results demonstrate successful sim-to-real transfer and validate the practical deployment readiness of our VLA approach for autonomous UAV reasoning navigation.

## A.7 LIMITATIONS AND FUTURE WORK

Our methodology exhibits two primary constraints that limit its applicability to complex real-world scenarios. First, AutoFly demonstrates limited global exploration capabilities in large-scale environments where targets may not be immediately visible within the camera's field of view, requiring systematic search strategies beyond our current reactive navigation approach. Second, our reliance on RGB imagery and monocular depth estimation, while sufficient for local obstacle avoidance and spatial reasoning, provides limited environmental context compared to 360° sensing modalities, potentially missing obstacles or features outside the forward-facing camera's scope. To address these limitations, we plan to enhance AutoFly's sensing capabilities through LiDAR integration, which will provide comprehensive 360° environmental perception and improve robustness in complex outdoor scenarios. While our dataset incorporates both static and dynamic obstacles during training and evaluation, the current Supervised Fine-Tuning (SFT) paradigm lacks direct environmental interaction, limiting the model's ability to develop fully adaptive strategies for highly dynamic real-world scenarios. Future work will integrate Reinforcement Learning to enable active interaction with dynamic environments, allowing the system to learn more robust reactive behaviors through trial-and-error exploration.

