# OpenReview forum: "AutoFly: Vision-Language-Action Model for UAV Autonomous Navigation in the Wild"
_ICLR.cc/2026/Conference — ICLR 2026 Poster_

### Official Review · Reviewer_hBdj · 2025-10-25

**Soundness:** 3
**Presentation:** 3
**Contribution:** 3
**Rating:** 6
**Confidence:** 5

**Summary:**

This paper proposes AutoFly, a vision-language-action (VLA) model for aerial vision-language navigation.  Without the need for detailed language instructions, AutoFly takes as input coarse-grained positional or directional guidance and RGB observations, and predict continuous-value velocity action control.  Specifically, AutoFly utilizes off-the-shelf monocular depth estimator to predict pseudo depth maps, followed by depth encoder, to facilitation 3D visual understanding.  In contrast to existing datasets that heavily depend on detailed instructions and unrealistic trajectory annotations, this paper proposes a new dataset based on AirSim, where (1) expert demonstrations are generated via RL, and (2) long-horizon trajectorizes are automatically segmented to distinguish avoidance and target-seeking phases.  Training with human-labeled trajectories, AutoFly demonstrates strong sim-to-real generalization while outperforming other VLA baselines in simulation by a large margin.

**Strengths:**

1. This paper is well written.  It clearly illustrates the limitations of prior args and the implementation details of the proposed method.
2. This paper demonstrates the effectiveness of the proposed method with strong sim2real generalization.
3. Aside from task success rate and path efficiency rate, this paper adopts an additional metric--collision rate, highlighting the importance of safety when deploying physical agents in the real world.
4. This paper presents detailed ablation study, demonstrating the efficacy of each model component.

**Weaknesses:**

1. This paper lacks the sim2real evaluation of previous simulation benchmark.  In the introduction, the authors state that "These limitations manifest in two critical areas: ..., and (2) insufficient real-world data representation, creating a significant sim-to-real gap. To bridge this gap, we present a novel autonomous navigation dataset specifically designed to address each identified limitation".  However, this paper does not showcase the sim2real gap of existing datasets in the experiments.  Without support from experimental results, I'd consider this as overclaimed statement.
2. This paper lacks discussion of recent studies on training-free aerial navigation policies, which also do not required detailed instructions.  For example, See-Point-Fly [1] suggests that navigation tasks are inherent visual grounding tasks in static scenes. Since VLMs excel at visual grounding, one can directly use VLM models to generate 2D waypoints for 3D navigation, without the need for training navigation policies. I'd strongly encourage the authors to include discussions on these new perspectives.
3. The newly introduced dataset only consider static scenes, while the real-world environments are often dynamic.  I'd highly encourage the authors to include discussion of static vs. dyanmic scenes in the limitation section.

---

Reference

[1] Hu, Chih Yao, et al. "See, Point, Fly: A Learning-Free VLM Framework for Universal Unmanned Aerial Navigation." Conference on Robot Learning. PMLR, 2025.

**Questions:**

1. What's the sim2real performance of AutoFly trained on existing navigation dataset?
2. Can you discuss recent studies that reformulate navigation tasks as visual grounding tasks?
3. Can you discuss the limitations of static scenes adopted in the proposed dataset?

---

> ### Author Response · Authors · 2025-11-22
> **The Response to Reviewer hBdj (Part 1/3)**
>
> We thank reviewer hBdj for the valuable time and constructive feedback.
>
> ---
>
> **Q1: About sim2real gap for previous simulation study**
>
> A1: We appreciate the reviewer's comment.
>
> We respectfully note that previous UAV VLN datasets are predominantly purely simulation-based. Consequently, the significant sim-to-real gap in Vision-Language Navigation (VLN) is a well-established challenge that has been extensively discussed in prior work. Existing studies have identified three primary causes for this gap[1-2]:
>
> 1. Visual Domain Gap: The texture and lighting differences between simulation environments and the real world.
> 2. Sensor Noise: The inevitable noise and artifacts in physical camera and sensor readings.
> 3. Cumulative Controller Errors: The drift and execution errors inherent in physical flight controllers compared to ideal simulators.
>
> Furthermore, our comparative experiments (detailed in Response Q4) demonstrate that models trained on existing simulation datasets fail completely in real-world scenarios (100% collision rate). This empirical evidence strongly corroborates that the aforementioned factors, including visual domain gap, sensor noise, and cumulative controller errors are indeed the critical bottlenecks preventing successful sim-to-real transfer for existing methods, resulting in the great gap.
>
> [1]. Anderson P, Shrivastava A, Truong J, et al. Sim-to-real transfer for vision-and-language navigation[C]//Conference on Robot Learning. PMLR, 2021: 671-681.
>
> [2]. Coursey A, Quinones-Grueiro M, Biswas G. Quantifying the Sim-To-Real Gap in UAV Disturbance Rejection[C]//35th International Conference on Principles of Diagnosis and Resilient Systems (DX 2024). Schloss Dagstuhl–Leibniz-Zentrum für Informatik, 2024: 16: 1-16: 18.

---

> ### Author Response · Authors · 2025-11-22
> **The Response to Reviewer hBdj (Part 2/3)**
>
> ---
>
> **Q2: About discussions on new perspectives like SPF**
>
> A2: We appreciate the reviewer's insightful suggestion to discuss training-free navigation paradigms (e.g., SPF). According to the reviewers' comments, we have also discussed this part of the content in the revision, which can be found in Line 111.
>
> **Paradigm shift and its value:** Recent work has introduced an innovative perspective by reformulating UAV navigation as a visual grounding problem. Methods like SPF leverage the exceptional zero-shot visual grounding capabilities of large-scale pre-trained VLMs to directly generate 2D waypoints from natural language descriptions, then use geometric projection to derive 3D navigation commands. This paradigm offers several compelling advantages:
>
> - Zero-shot generalization: By operating directly on foundation models trained on internet-scale data, these approaches can handle novel objects and scenes without task-specific training.
> - Natural sim-to-real transfer: Since they work directly on real-world RGB images without requiring simulated training data, they inherently avoid the visual domain gap that plagues learning-based methods.
> - Interpretability: The explicit waypoint generation provides clear visualization of the system's intent, facilitating debugging and user trust.
>
> We agree that this is a highly promising direction that opens new possibilities for deploying UAV systems in open-world scenarios.
>
> **Critical constraints for agile UAV navigation:** However, through rigorous experimental evaluation, we identify fundamental limitations when applying these training-free methods to our task setting—continuous autonomous navigation requiring integrated obstacle avoidance, path planning, and target recognition in dense, obstacle-rich environments. To provide concrete evidence, we conducted comparative experiments with SPF under identical conditions:
>
> | Method | SR (Real-world) | CR (Real-world) | PER (Real-world) |
> | --- | --- | --- | --- |
> | AutoFly | 60% | 30% | 76.5% |
> | SPF | 10% | 85% | 45.1% |
>
> | Method | SR (Simulation) | CR (Simulation) | PER (Simulation) |
> | --- | --- | --- | --- |
> | AutoFly | 47.9% | 21.9% | 77.3% |
> | SPF | 27.8% | 75.0% | 55.1% |
>
> The dramatic performance gap reveals two critical bottlenecks:
>
> 1. Continuous Planning in Complex Environments: General-purpose VLMs excel at high-level visual grounding in relatively sparse scenes, but lack the specific fine-tuning required for continuous, granular trajectory planning in obstacle-dense environments. Our environments demand rapid, reactive control sequences for safe navigation through complex obstacle configurations. SPF's hierarchical waypoint-based approach struggles to generate sufficiently dense and precise waypoints at the frequency required for agile flight, leading to the extremely high collision rate (85% in real-world, 75% in simulation). In contrast, AutoFly's end-to-end fine-tuning on dense trajectory data enables continuous, smooth control outputs optimized for collision avoidance.
> 2. Real-World Latency and Stability: Physical UAV deployment is highly sensitive to inference latency. During real-world testing, we observed that SPF's reliance on large language model inference frequently exceeds 150ms per decision (excluding communication overhead), preventing real-time reactive control. This latency is catastrophic when navigating through dynamic or rapidly approaching obstacles, where sub-100ms response times are often required. In contrast, our optimized deployment pipeline (TensorRT-LLM, model parallelism) achieves stable 15 FPS (∼67ms per frame) with low variance, ensuring reliable real-time decision-making capability essential for flight safety.
>
> **Complementary perspectives:** We emphasize that these findings do not diminish the value of training-free approaches. Rather, they highlight the complementary nature of the two paradigms:
>
> - Training-free methods (e.g., SPF): Excel in open-world scenarios requiring semantic understanding of diverse, previously unseen objects. They are particularly well-suited for high-level navigation tasks in relatively sparse environments where waypoint-based planning suffices (e.g., "fly to the nearest red car in a parking lot").
> - Fine-tuned VLA methods (AutoFly): Excel in safety-critical, high-frequency control tasks requiring continuous obstacle avoidance and precise trajectory execution. They are essential for navigating dense, complex environments where reactive control is paramount.
>
> Future research could explore hybrid architectures that combine the open-world generalization of training-free VLMs for high-level goal specification with the precise, low-latency control of fine-tuned VLA models for safe trajectory execution.

---

> ### Author Response · Authors · 2025-11-22
> **The Response to Reviewer hBdj (Part 3/3)**
>
> **Q3: About limitations of static scenes**
>
> A3: We appreciate the reviewer's valuable suggestion regarding static vs. dynamic scenes. According to the reviewers' comments, we have also discussed this part of the content in the revision, which can be found in Line 1167.
>
> **Mixed Static-Dynamic Scenarios:** We clarify that our experimental setup is not limited to static scenes. In fact, both our training and testing environments explicitly incorporate a combination of static and dynamic elements. During data collection, we deliberately configured a subset of environments with moving obstacles to enrich trajectory diversity, thereby better approximating real-world dynamics.
>
> **Limitations and Future Work:** We agree with the reviewer that real-world environments are inherently dynamic. While our dataset design addresses this partially, we acknowledge that the current Supervised Fine-Tuning (SFT) paradigm lacks direct, active interaction with the environment during training. To bridge this gap, we plan to incorporate Reinforcement Learning (RL) in future work. This will enable the model to learn adaptive strategies through direct interaction with dynamic environments, further enhancing its robustness. We explicitly discuss this static-dynamic distinction and the potential of RL in the Limitations section of our revised manuscript.
>
> ---
>
> **Q4: About sim-to-real performance for AutoFly trained on existing navigation dataset**
>
> A4: We appreciate the reviewer's suggestion.
>
> To rigorously evaluate this, we trained AutoFly using the OpenUAV dataset, a representative existing UAV navigation benchmark. To ensure a fair comparison, we aligned our model's action space with OpenUAV's position control interface (converting from our native velocity control) and carefully filtered the observation data. We then conducted real-world outdoor testing (20 trials) using this model.
>
> The results showed a 100% collision rate (0% success rate). This complete failure in real-world transfer can be attributed to two critical factors:
>
> 1. Visual Domain Gap: The visual fidelity of OpenUAV's simulation data differs significantly from real-world outdoor scenes, leading to catastrophic perception failures.
> 2. Controller Error Accumulation: OpenUAV relies on position control, which assumes perfect odometry in simulation. In the real world, visual odometry drift and accumulated controller errors cause the UAV to deviate significantly from planned paths, a problem our velocity-based approach mitigates but position-based training exacerbates.
>
> These findings validate our claim that existing datasets and paradigms exhibit a great gap for robust sim-to-real transfer, necessitating the high-fidelity data and approach proposed in AutoFly.
>
> ---
>
> **Q5: About visual grounding navigation tasks**
>
> A5: Thanks for your comment!
>
> Recent advances[1-3] have significantly reformulated navigation tasks from end-to-end action prediction to visual grounding and reasoning problems, leveraging the open-vocabulary capabilities of Vision-Language Models (VLMs). Instead of directly mapping observations to discrete actions, these approaches adopt a hierarchical paradigm: they first ground linguistic instructions onto specific visual regions, such as semantic maps, 2D waypoints, or future keyframes and subsequently derive control actions through geometric projection or classic planners.
>
> This formulation fundamentally shifts the challenge from learning control policies to spatial-semantic understanding. By decoupling high-level reasoning from low-level execution, these methods bypass the need for large-scale domain-specific action datasets and offer superior interpretability. They allow the system to explicitly visualize where it intends to go before moving, directly harnessing the zero-shot generalization capabilities of foundation models trained on internet-scale data.
>
> [1]. Chu M, Zheng Z, Ji W, et al. Towards natural language-guided drones: GeoText-1652 benchmark with spatial relation matching[C]//European Conference on Computer Vision. Cham: Springer Nature Switzerland, 2024: 213-231.
>
> [2]. Liu Q, Zhang S, Qiao Y, et al. GroundingMate: Aiding Object Grounding for Goal-Oriented Vision-and-Language Navigation[C]//2025 IEEE/CVF Winter Conference on Applications of Computer Vision (WACV). IEEE, 2025: 1775-1784.
>
> [3]. Yokoyama N, Ha S, Batra D, et al. Vlfm: Vision-language frontier maps for zero-shot semantic navigation[C]//2024 IEEE International Conference on Robotics and Automation (ICRA). IEEE, 2024: 42-48.

---

> > ### Comment · Reviewer_hBdj · 2025-11-23
> >
> > Thank the authors for the thorough response.  My concerns have been addressed.

---

> > > ### Author Response · Authors · 2025-11-23
> > > **The Response to Reviewer hBdj**
> > >
> > > We sincerely thank you for your response. Please let us know if you have further questions.  As promised in our rebuttal, we will ensure that these points are thoroughly reflected in the final revised version of the manuscript.  Thank you again for your insightful comments, which have significantly helped in improving our work.

---

### Official Review · Reviewer_Hzvn · 2025-10-27

**Soundness:** 3
**Presentation:** 3
**Contribution:** 3
**Rating:** 6
**Confidence:** 3

**Summary:**

AutoFly presents a vision-language-action model for autonomous UAV navigation using coarse directional guidance rather than detailed instructions. It employs a pseudo-depth encoder (Depth Anything V2) with Siamese MLP projectors for depth-visual alignment, achieving 47.9% simulation success rate (3.9% over OpenVLA) and 60% real-world success on a new 13K+ episode dataset. Strengths include strong problem motivation, comprehensive ablation studies, and real-world validation. Concerns: modest gains given the added complexity, a low obstacle encounter rate (10 avg vs. 83 in AerialVLN), suggesting easier scenarios; heavy simulation dependence (success drops 60%->10% with less sim data); and unclear depth-generation overhead affecting the 15 FPS inference rate.

**Strengths:**

- The shift from detailed instruction-following VLN to autonomous navigation with coarse guidance addresses a real deployment gap.
- Using monocular depth estimation (Depth Anything V2) instead of depth sensors is elegant and practical. It avoids sim-to-real depth sensor gaps while adding spatial reasoning capabilities with only RGB cameras.
- The shared-weight design for depth and visual token projection is simple yet effective, enforcing consistent cross-modal representations.

**Weaknesses:**

- The 3.9% improvement in success rate over OpenVLA (47.9% vs 44%) is modest, given the additional pseudo-depth encoder, depth generator, and specialized projectors. The paper lacks analysis to show whether simpler approaches (e.g., better vision encoders or data augmentation) could achieve similar gains without architectural changes.
- The average obstacle encounter rate of 10 is significantly lower than comparable UAV datasets (AerialVLN: 83, OpenUAV: 104, CityNav: 26). This contradicts the paper's framing of "autonomous navigation in the wild" and "dense, obstacle-rich environments."
- Only 200 real-world trials in "reconstructed real environments" (suggesting indoor lab settings) do not validate outdoor autonomous navigation. The dramatic performance drop from 60% (10K sim + 1K real) to 10% (0K sim + 1K real) reveals poor sim-to-real transfer and heavy dependence on simulation data.
- The paper only compares against manipulation-focused VLA models (RT-1, RT-2, OpenVLA), which are not designed for UAV navigation. There are no comparisons to established autonomous UAV navigation methods, classical planning algorithms combined with learned perception, or recent outdoor navigation works (e.g., the cited Zhou et al. 2021a,b; Xu et al. 2022, and not cited SPF [A] and OpenFly [B]).

[A] Hu, Chih Yao, et al. "See, Point, Fly: A Learning-Free VLM Framework for Universal Unmanned Aerial Navigation." Conference on Robot Learning. PMLR, 2025.

[B] Gao, Yunpeng, et al. "OpenFly: A Comprehensive Platform for Aerial Vision-Language Navigation." arXiv preprint arXiv:2502.18041 (2025).

**Questions:**

- Can you provide ablation studies isolating the contribution of depth information from other factors (e.g., additional training data, different augmentations, and architectural choices)? What is the success rate when using the same dataset without the pseudo-depth encoder?
- What is the inference time breakdown for each component (Depth Anything V2, visual encoder, depth projector, LLM)? How does the 15 FPS compare to OpenVLA on the same hardware? Is real-time performance achievable without the extensive optimization pipeline (TensorRT, ONNX conversion, model parallelism)?
- Why is your average obstacle encounter rate (10) so much lower than comparable UAV datasets? Does this metric measure total obstacles in the environment or actual collision-risk encounters during flight? Can you provide comparisons of trajectory length and obstacle density between AerialVLN and CityNav?

---

> ### Author Response · Authors · 2025-11-22
> **The Response to Reviewer Hzvn (Part 1/3)**
>
> We thank reviewer Hzvn for the valuable time and constructive feedback.
>
> ---
>
> **Q1: About simpler approach modifications gain**
>
> A1: We appreciate the reviewer's suggestion to verify if simpler modifications could achieve similar gains.
>
> **Baseline Stability:** First, we emphasize that our large-scale test set ensures high statistical power with very low variance in success rates. Consequently, the 3.9% improvement provided by the depth encoder represents a substantial and robust performance gain.
>
> **Comparison with Simpler Approaches:** To rigorously test if this gain could be achieved without the depth module, we conducted three additional experiments:
>
> 1. Scaling Data: We collected an additional 1,000 episodes (approximately 350,000 Vision-Language-Action pairs) to test the effect of data scaling.
> 2. Data Augmentation: We applied photometric augmentations (random brightness, contrast, saturation, and hue). Geometric augmentations were excluded to preserve the alignment of action labels.
> 3. Stronger Vision Encoder: We upgraded the visual backbone by replacing SigLIP with the state-of-the-art SigLIP 2 (fused with DINOv2) to see if a better RGB encoder eliminates the need for explicit depth.
>
> **Results:** As shown in the table below, while adding training data provided the largest boost among these baselines (+1.2%), it still falls significantly short of the +3.9% gain achieved by our pseudo-depth encoder (47.9% SR). This confirms that the specific geometric and spatial reasoning capabilities introduced by the depth encoder are essential for this task and cannot be replicated merely by scaling data or upgrading the RGB perception features.
>
> | Method | SR | CR | PER |
> | --- | --- | --- | --- |
> | Baseline (OpenVLA) | 44.0 | 24.5 | 75.1 |
> | Data Augmentations | 44.3 (+0.3) | 24.1 | 75.5 |
> | Stronger Encoder (SigLIP 2) | 44.7 (+0.7) | 23.1 | 76.6 |
> | Add Training Data (1k eps) | 45.2 (+1.2) | 23.6 | 76.5 |
> | AutoFly (w/ Depth Encoder) | 47.9 (+3.9) | 21.9 | 77.3 |
>
> ---
>
> **Q2: About obstacle encounter in dataset**
>
> A2: Thanks for your great comment.
>
> **Clarification on obstacle density:** We respectfully point out a misunderstanding regarding the metrics in Table 1. The values cited by the reviewer (AerialVLN: 83, OpenUAV: 104, CityNav: 26) correspond to Instruction Length (Column 6), not the number of obstacles.
>
> In fact, the average number of obstacles encountered in our dataset (10) is greater than or equal to those scenes in authoritative navigation work[1][2].
>
> **Metric measurement:** Our "average obstacle encounter number" (column 7) measures the number of obstacles the UAV encounters along its trajectory from start to goal that pose actual collision risks. Higher obstacle encounters directly correlate with higher collision risk during navigation.
>
> **Comparisons between AerialVLN and CityNav:** As shown in Table 1 (Column 5), we explicitly compared the average trajectory lengths of our dataset with AerialVLN, CityNav, and other authoritative datasets. Our trajectories maintain lengths comparable to these benchmarks.
>
> Datasets like AerialVLN, CityNav focus primarily on vision-and-language navigation, following pre-defined language instructions rather than autonomous obstacle avoidance and planning. Consequently, they do not typically report obstacle encounter statistics. To verify this difference, we visualized the simulation environments of AerialVLN and CityNav. Our analysis reveals that during instruction-following flight trajectories in these datasets, obstacles are essentially absent, with only sparse isolated scenes that contain 1-2 obstacles that do not materially impact flight paths. This fundamentally differs from our task design, which requires continuous obstacle avoidance planning integrated with object recognition.
>
> [1]. Zhang Y, Hu Y, Song Y, et al. Learning vision-based agile flight via differentiable physics[J]. Nature Machine Intelligence, 2025: 1-13.
>
> [2]. Ren Y, Zhu F, Lu G, et al. Safety-assured high-speed navigation for MAVs[J]. Science Robotics, 2025, 10(98): eado6187.

---

> ### Author Response · Authors · 2025-11-22
> **The Response to Reviewer Hzvn (Part 2/3)**
>
> **Q3: Add the experiments in the wild**
>
> A3: Thanks for your comment.
>
> **Outdoor validation:** In response to the reviewer's concern about limited outdoor testing, we conducted additional outdoor experiments in a campus forest environment featuring naturally irregular trees, dynamic swaying branches, and unstructured vegetation. As shown in the table below, AutoFly achieves a 55% success rate in this outdoor environment, closely matching the 60% indoor performance. This minimal performance gap (only 5 percentage points) demonstrates AutoFly's robust capability for outdoor autonomous navigation and strong environmental adaptability across diverse settings. We provide a real flight demo in the wild, and the link is as follows:[ https://anonymous.4open.science/r/AutoFly-outdoor-75E5 ]. According to the reviewers' comments, we have added this part of the experiment to the revision version, which is located in Line 396 and 439.
>
> | scene | SR | CR | PER |
> | --- | --- | --- | --- |
> | indoor | 60(12/20) | 30 | 76.5 |
> | outdoor | 55(11/20) | 35 | 75.1 |
>
> **Sim-to-real transfer clarification:** We believe the reviewer may have misinterpreted the experimental comparison. The performance difference does not indicate poor sim-to-real transfer or problematic dependence on simulation data. Rather, it demonstrates the effectiveness of simulation data for addressing real-world data scarcity.
>
> Data scale clarification: The notation refers to episode counts:
>
> - 10K sim + 1K real: 10,000 simulation episodes (2.5M vision-language-action pairs) + 1,000 real-world episodes (200K vision-language-action pairs) → 60% success rate
> - 0K sim + 1K real: Only 1,000 real-world episodes (200K vision-language-action pairs) → 10% success rate
>
> When real-world data is limited (only 1K episodes), co-training with large-scale simulation data dramatically improves performance (from 10% to 60%), demonstrating effective sim-to-real transfer. In principle, with sufficient real-world data, the model would not require simulation data. However, collecting large-scale real-world UAV navigation data is expensive, dangerous, and time-consuming. Our results validate that simulation-augmented training is an effective and practical strategy for achieving strong real-world performance with limited real data, precisely the scenario most relevant for practical deployment.
>
> ---
>
> **Q4: Comparisons with recent navigation**
>
> A4: Thanks for your professional comment.
>
> **Rationale for baseline selection:** Our task requires the UAV to continuously perform obstacle avoidance planning while simultaneously locating targets specified by coarse-grained language commands. This fundamentally differs from existing method categories:
>
> - Classical planning algorithms: Methods focus primarily on path planning and obstacle avoidance without language input or target recognition capabilities
> - Classical UAV-VLN methods: Methods (e.g., the cited works) focus on instruction-following navigation where the UAV follows detailed step-by-step language descriptions, without autonomous obstacle avoidance planning
>
> Since our approach is designed within the VLA (Vision-Language-Action) model paradigm, we selected three representative VLA architectures as baselines to ensure fair comparison:
>
> - RT-1: Pre-VLM baseline architecture
> - RT-2: VLM-based architecture
> - OpenVLA: State-of-the-art VLM-based baseline
>
> This design choice explains why we did not compare against classical planning algorithms or UAV-VLN methods: they address fundamentally different tasks.
>
> **SPF comparison:** In response to the reviewer's suggestion, we conducted additional comparisons with SPF under identical conditions (both simulation and real-world environments). As shown in the tables, SPF significantly underperforms AutoFly across both settings:
>
> | method | SR | CR | PER |
> | --- | --- | --- | --- |
> | AutoFly | 60(12/20) | 30 | 76.5 |
> | SPF | 10(2/20) | 85 | 45.1 |
>
> | method | SR | CR | PER |
> | --- | --- | --- | --- |
> | AutoFly | 47.9 | 21.9 | 77.3 |
> | SPF | 27.8 | 75 | 55.1 |
>
> **Performance analysis:** SPF's poor performance stems from two factors:
>
> 1. Training-free limitation: As a training-free method, SPF lacks task-specific fine-tuning. Our environments require continuous obstacle avoidance planning, making them substantially more complex than the scenarios in SPF's original paper.
> 2. Real-time deployment challenges: During real-world deployment, we observed that SPF's large language model inference latency frequently exceeds 150ms (excluding communication overhead), preventing real-time decision-making. This latency is catastrophic for obstacle avoidance planning where timely reactions are critical.
>
> These results validate that in complex environments requiring integrated planning, avoidance, and recognition, AutoFly's fine-tuned approach achieves superior autonomous navigation performance, while our deployment optimization pipeline ensures real-time decision-making capability on physical UAV platforms.

---

> ### Author Response · Authors · 2025-11-22
> **The Response to Reviewer Hzvn (Part 3/3)**
>
> **Q5: About deployment**
>
> A5: Thanks for your comment.
>
> **Inference time breakdown:** We provide detailed per-component inference times on NVIDIA RTX 4090 hardware:
>
> | components | precision | inference times |
> | --- | --- | --- |
> | SigLIP  | FP16 | 3-5ms |
> | DinoV2 | FP16 | 3-6ms |
> | Depth Anything V2 | FP16  | 11-13ms |
> | Other kernels and projectors | BF16 | <1ms |
> | LLM with TensorRT-LLM | BF16 | 50-70ms |
>
> **Comparison with OpenVLA:** AutoFly achieves inference speeds comparable to OpenVLA (40ms-60ms), with only a marginal increase in latency. This slight difference stems primarily from the LLM inference stage, where processing the additional depth embeddings adds a small amount of computational load. Crucially, however, the latency of the depth encoder itself (11-13ms) is effectively hidden by our multi-process parallel architecture, which pipelines depth extraction to execute concurrently with the dominant LLM inference (50-70ms). This design ensures that adding explicit depth perception incurs minimal overall overhead.
>
> **Necessity of optimization pipeline:** Real-time performance is not achievable without the extensive optimization pipeline. Without TensorRT, ONNX conversion, and model parallelism optimizations, the system achieves only 3 FPS on RTX 4090, far below the real-time threshold required for UAV navigation. The optimization pipeline is therefore essential, not optional, for practical deployment.

---

> > ### Comment · Reviewer_Hzvn · 2025-11-23
> > **thank you**
> >
> > Thank the authors for the thorough rebuttal. The rebuttal resolves most of my concerns regarding the novelty, experimental rigor, and dataset complexity. Specifically, I appreciate the authors conducting comparisons with SigLIP 2 and SPF within one week. These comparisons make the contribution much clearer. I would like to raise my score to Accept. I also recommend that the authors include these new ablation results and the additional comparisons with SigLIP 2 and SPF in the revised paper.

---

> > > ### Author Response · Authors · 2025-11-23
> > > **The Response to Reviewer Hzvn**
> > >
> > > We sincerely thank you for your response. We will include these new ablation results and the additional comparisons with SigLIP 2 and SPF in the revised paper followed by your suggestions. Please let us know if you have further questions. Once again, we extend our sincere appreciation for your valuable feedback and the opportunity you have afforded us throughout the review process.

---

### Official Review · Reviewer_ytdu · 2025-10-28

**Soundness:** 3
**Presentation:** 2
**Contribution:** 4
**Rating:** 6
**Confidence:** 4

**Summary:**

This paper introduces AutoFly, an end-to-end Vision-Language-Action (VLA) model designed to enable Unmanned Aerial Vehicles (UAVs) to navigate autonomously in unknown environments using only coarse, high-level linguistic instructions. This approach addresses a key limitation of current Vision-Language Navigation (VLN) systems, which rely on detailed, step-by-step instructions and predetermined routes that are unavailable in real-world scenarios. AutoFly enhances its spatial reasoning by incorporating a pseudo-depth encoder that generates depth-aware features from standard RGB camera inputs. To support this autonomous paradigm, the authors also developed a new dataset focused on continuous obstacle avoidance and planning rather than simple instruction-following, which includes extensive real-world trajectories to bridge the simulation-to-reality gap.

**Strengths:**

1. The paper correctly identifies a significant limitation in existing UAV VLN research: an over-reliance on detailed, step-by-step instructions that are often unavailable in real-world, unknown environments. The proposed shift to a paradigm using only coarse directional guidance is a practical and valuable step toward more robust, autonomous agents that can operate with minimal human guidance.
2. The authors recognize that existing datasets are ill-suited for this new, autonomous navigation task. A major strength of this work is the construction of a new, large-scale autonomous navigation dataset, comprising over 13000 trajectories. Crucially, the dataset also includes 1000 real-world flight episodes to help bridge the sim-to-real gap.
3. The paper introduces a novel "pseudo-depth encoder" to derive depth-aware spatial features directly from monocular RGB inputs. This directly addresses the critical need for 3D geometric understanding in UAV navigation, a known limitation of RGB-only systems. The reason for using a pseudo-depth generator instead of idealized simulator depth or specialized hardware is well-argued, as it aims to improve sim-to-real transfer and reduce the payload and cost of the final UAV platform. This architectural choice is validated by an ablation study showing a 3.9% improvement in success rate.

**Weaknesses:**

1. The title and abstract promise autonomous navigation "in the wild". However, the real-world experiments are limited in scope and do not support this claim.
The paper states real-world data is acquired "within controlled laboratory environments".
The visualization of the real-world test in Figure 5, Figure 15 clearly shows a structured, indoor lab setting, not a dynamic "wild" environment.
2. The model's formulation defines the policy as taking only the current RGB observation $o_t$ as input, along with the language instruction. This makes the model fundamentally memoryless. It has no mechanism to build a persistent understanding of the environment, remember areas it has explored, or recall the location of obstacles it has passed. This is a severe limitation for any non-trivial navigation task, especially in "wild" environments where a target may be occluded, requiring the agent to remember its location from previous observations and go around obstacles.
3. I think there is major unclarity in the core model architecture, particularly the vision encoder. The main methodology states the model is initialized with a "prism-siglip-7b" configuration and in section 3.2, “Action De‑tokenizer,” Eq. 1 shows the model indeed only uses the SigLIP vision encoder. However, the main results in Table 2 report a 47.9% success rate (SR) and the ablation study in the Appendix (Figure 11) shows that a "SigLIP" only encoder achieves a 46.6% SR , while a "DINO-SigLIP" fusion achieves the 47.9% SR, the same as main results. This implies the final AutoFly model does use a DINO-SigLIP fusion for its visual encoder. Moreover, in section 4.4 the Table 4 shows that removing the pseudo-depth encoder drops the SR from 47.9% to 44.0%. These 44.0% SR, 24.5% CR and 75.1% PER results are the same as the OpenVLA baseline which can mean that your model is the OpenVLA with additional depth encoder.

**Questions:**

1. Your model's policy is formulated as $\pi(a_t | o_t, L)$, conditioned only on the current RGB observation $o_t$. This makes the agent purely without memory. Why is no visual/action history modeled? How can this architecture handle non-trivial navigation where the target is temporarily occluded by an obstacle?
2. Please try to address the model architecture concerns from the Weaknesses section. As mentioned, I think your paper's core architecture is contradictory. Please clarify the vision encoder used in the final AutoFly model: Eq. (1) shows the LLM consuming SigLIP visual embeddings (plus depth), whereas the best ablation uses a DINO–SigLIP fusion that matches your main 47.9% SR score.
3. Table 4 shows that removing the pseudo-depth encoder yields the same numbers as your OpenVLA baseline in Table 2. Does your model without the pseudo-depth encoder reduce to OpenVLA?
4. The AutoFly training process is note clear. It is mentioned that in Stage 2 you jointly fine-tuning the pseudo-depth encoder alongside the pre-trained VLA backbone. However, in training details of section 4.1 you didn't mention the learning rate of pseudo-depth generator, only pseudo-depth projector and VLM. Is the pseudo-depth generator (Depth Anything V2) frozen?

---

> ### Author Response · Authors · 2025-11-22
> **The Response to Reviewer ytdu (Part 1/2)**
>
> We thank reviewer ytdu for the valuable time and constructive feedback.
>
> ---
>
> **Q1: Add the experiments in the wild**
>
> A1: Thank you for your insightful and constructive comments.
>
> We clarify our experimental paradigm: All simulation environments are designed to represent outdoor, unstructured "wild" scenarios with irregular obstacles. For the real-world deployment experiments, safety considerations necessitated conducting initial tests indoors; however, these indoor environments were deliberately configured to replicate outdoor unstructured scenes with irregular obstacles, varied object placements, and challenging navigation conditions.
>
> In response to the reviewer's concern, we conduct additional outdoor experiments to validate AutoFly's performance beyond controlled laboratory settings. The outdoor experiments are performed in a campus forest environment featuring naturally irregular trees, dynamic swaying branches, and unstructured vegetation, conditions representative of less controlled outdoor scenarios. As shown in the table below, AutoFly achieves a 55% success rate in this outdoor environment, closely matching the 60% indoor performance. This minimal performance gap (only 5 %) demonstrates AutoFly's robust capability for outdoor autonomous navigation and strong environmental adaptability across diverse settings. The comparable collision rates (30% indoor vs. 35% outdoor) and path efficiency (76.5% vs. 75.1%) further validate the system's consistent performance regardless of environmental conditions, including the presence of dynamic natural elements such as moving branches. We provide a real flight demo in the wild, and the link is as follows:[ https://anonymous.4open.science/r/AutoFly-outdoor-75E5 ]. According to the reviewers' comments, we have added this part of the experiment to the revision version, which is located in Line 396 and 439.
>
> | scene | SR | CR | PER |
> | --- | --- | --- | --- |
> | indoor | 60(12/20) | 30 | 76.5 |
> | outdoor | 55(11/20) | 35 | 75.1 |
>
> ---
>
> **Q2: About visual/action history modeled**
>
> A2: Thanks for your good comment.
>
> **Visual/action history modeled:** We acknowledge this as a limitation of the current design. Our decision to exclude explicit history modeling was driven primarily by the need for computational efficiency and real-time system responsiveness, which are critical for practical UAV deployment. We employ a streaming pipeline architecture specifically designed to support multi-process parallel acceleration. This design choice minimizes computational overhead and latency, ensuring the system is real-time for real-world deployment. We provide the following specific analysis regarding the detailed reasons for not conducting visual and action modeling.
>
> - Visual History: We did not incorporate visual history modeling because the computational cost of extracting features across multiple frames is prohibitively high. Maintaining a single-frame inference model allows us to preserve the high frequency needed for agile flight.
> - Action History: Regarding action history, our control space is based on velocity commands, which ensures the UAV's agility in complex, continuous obstacle avoidance scenarios. Modeling velocity history provided negligible gains in environmental understanding or mapping capabilities compared to direct visual perception, and thus was omitted to streamline the architecture.
>
> **Handling temporary occlusions:** Regarding the reviewer's question about temporary target occlusion, we acknowledge that this is a fundamental challenge for memoryless architectures. While incorporating visual history modeling would be the ideal solution to address this issue, our pseudo-depth encoder can partially mitigate this limitation.
>
> The depth encoder enhances AutoFly's spatial reasoning capabilities, enabling the system to better understand the 3D spatial layout of the environment. When the target is temporarily occluded, the enhanced depth awareness allows the UAV to perform more effective local path planning around obstacles while maintaining general progress toward the target region.
>
> **Future directions:** We acknowledge that incorporating explicit memory mechanisms would further enhance performance. However, it is important to emphasize that this work establishes a baseline approach for the novel autonomous navigation task with coarse-grained instructions that a task paradigm that has not been previously explored in the UAV navigation literature. Our current memoryless architecture serves as a foundational baseline that demonstrates the feasibility of this challenging task setting, achieving 47.9% success rate even without temporal modeling.
>
> Building upon this baseline, we plan to investigate temporal modeling approaches in future work, including visual history encoding and action sequence modeling, to address the limitations identified by the reviewer. These extensions will explore how memory mechanisms can further improve performance.

---

> ### Author Response · Authors · 2025-11-22
> **The Response to Reviewer ytdu (Part 2/2)**
>
> **Q3: About  model architecture**
>
> A3: We sincerely apologize for the confusion caused by this inconsistency in our manuscript. The reviewer is correct to identify this discrepancy. The vision encoder used in our final AutoFly model is indeed the DINO–SigLIP fusion configuration, which corresponds to the best-performing ablation variant achieving the 47.9% success rate reported in our main results.
>
> We have corrected the equation 1 in the revision version to accurately reflect this DINO–SigLIP fusion architecture, ensuring consistency between our mathematical formulation and the experimental configuration. Thank you for bringing this important clarification to our attention.
>
> ---
>
> **Q4: Ablation study about pseudo-depth encoder**
>
> A4: We thank the reviewer for their good comments.
>
> **Equivalence to OpenVLA:** Since our final AutoFly model adopts the dual SigLIP-DINO visual encoder, removing the pseudo-depth encoder results in a LLaVA-based architecture that is structurally identical to the OpenVLA baseline. Consequently, the performance metrics that we used are the same as those of OpenVLA.
>
> **Reproducibility and Validation:** To verify this, we conducted two additional reproduction runs of the ablation setting (AutoFly w/o pseudo-depth encoder). As shown in the table below, the results exhibit minimal variance across runs (mean variance is negligible), confirming that referencing the baseline values is methodologically sound.
>
> This comparison isolates the specific contribution of our proposed module. The significant performance drop observed when removing the depth module (from 47.9% to 44.0%) confirms that the pseudo-depth encoder is the primary driver of the performance gains, as it substantially enhances the model's spatial reasoning and trajectory planning capabilities.
>
> | group | SR | CR | PER |
> | --- | --- | --- | --- |
> | 1 | 43.0 | 25.7 | 74.4 |
> | 2 | 44.0  | 24.5  | 75.1 |
> | 3 | 43.3 | 23.6 | 75.9 |
> |  | 43.4 ± 0.5 | 24.6 ± 1.1 | 75.1 ± 0.8 |
>
> ---
>
> **Q5: Two-stage training paradigm details**
>
> A5:We provide detailed training specifications for our two-stage training paradigm:
>
> **Stage 1 (Vision-Language Alignment):** We follow the configuration of Prismatic-VLMs' prism-dinosiglip+7b model, using its pretrained weights to initialize the MLP Projector and LLM. For comprehensive training details of this stage, we refer readers to the Prismatic-VLMs implementation.
>
> **Stage 2 (Robot Action Fine-tuning):** We freeze the Depth Generator component within the Pseudo Depth Encoder while training the following modules: Depth Projector, MLP Projector, LLM, and Visual Encoder. The hyperparameters are configured as follows:
>
> - Learning rate: 1e-4 for the Depth Projector; 2e-5 for all other trainable modules
> - Scheduler: Constant
> - Optimizer: AdamW
> - Training steps: 80,000 steps (approximately 8 epochs)
> - Batch size: 32
> - Hardware: 8 NVIDIA A100 GPUs 80G

---

> > ### Comment · Reviewer_ytdu · 2025-11-26
> >
> > Thank you for the clarifications. My concerns have been addressed.

---

> > > ### Author Response · Authors · 2025-11-27
> > > **Thank you**
> > >
> > > Thank you very much for your response and insight feedback! Please let us know if you have further questions. We will further refine and polish the manuscript according to your suggestions to make it as strong as possible.

---

### Official Review · Reviewer_nXwK · 2025-10-29

**Soundness:** 3
**Presentation:** 2
**Contribution:** 3
**Rating:** 4
**Confidence:** 3

**Summary:**

This paper proposes a novel task and dataset that uses only coarse-grained instruction guidance to assist UAV navigation, and leverages  VLM and depth modal information to alleviate obstacle avoidance challenges in dynamic and unpredictable environments, achieving good performance in both simulation and real-world environments.

**Strengths:**

1. The novel integration of a pseudo-depth encoder. This enhances the model's geometric reasoning for obstacle avoidance and safe navigation without needing physical depth sensors, effectively bridging a critical gap for real-world deployment where detailed environmental data is unavailable.
2. The creation of a comprehensive autonomous navigation dataset. The dataset uniquely emphasizes real-world challenges like continuous obstacle avoidance and includes real flight data, facilitating robust sim-to-real transfer.
3. This paper conducted sufficient comparison and ablation experiments to prove that its method achieved excellent performance in both simulation and real environments.

**Weaknesses:**

1. VLM approach may cause computationally intensive over the more efficient reinforcement learning approach for obstacle avoidance and navigation. The authors should clarify why the VLM approach is irreplaceable over established RL methods when only obstacle avoidance and basic navigation are required.
2. The Pseudo-Depth Encoder only showed a 4% performance improvement in ablation study, raising doubts about the module's effectiveness. Could you provide examples of extreme scenarios such as dynamic obstacles to illustrate the module's robustness?
3. The initial position ${a_0}$ should be crucial for coarse navigation, but the method doesn't explain how it's used.
4. The two-stage training paradigm doesn't clearly explain how each stage is trained (which data is used, which module is fine-tuned, etc.) and the hyper-parameters such as epochs and learning rate.

**Questions:**

1. How is the L1-norm loss function described in Training Details used in the base language model?
2. What causes the task success rate to be only 50%-60%? The experiment needs to analyse more failure cases to explain the rationality of the success metric settings and the effectiveness of the depth encoder.
3. How does the instruction generated in the dataset?

---

> ### Author Response · Authors · 2025-11-22
> **The Response to Reviewer nXwK (Part 1/4)**
>
> We thank reviewer nXwK for the valuable time and constructive feedback.
>
> ---
>
> **Q1: Necessity of VLM**
>
> A1: Thanks for your careful review. We make detailed clarification below.
>
> **Clarifying differences in task complexity:** Our proposed autonomous flight task extends far beyond simple obstacle avoidance and basic navigation. The UAV must continuously perform obstacle avoidance and path planning while recognizing objects specified by coarse-grained language commands (e.g., "move forward and avoid obstacles to reach the red sports car"). This is a comprehensive navigation task that requires understanding natural language instructions while integrating obstacle avoidance, path planning and object recognition into a unified framework.
>
> Why VLM is essential for our task:
>
> - Vision-language alignment : The commands are natural language instructions, necessitating vision-language alignment capabilities that VLMs provide.
> - Multi-task integration: The integrated navigation task requires VLM's powerful generalization abilities to handle the complexity of simultaneous obstacle avoidance, planning and recognition.
>
> These capabilities are fundamentally unattainable using reinforcement learning models alone.
>
> **Necessity of VLM for basic navigation:** We acknowledge that for scenarios requiring only obstacle avoidance and basic navigation, compact RL-based models are more efficient. However, VLMs offer distinct advantages in areas where RL-based models face significant challenges:
>
> - Scene generalization: RL-based obstacle avoidance and basic navigation methods typically demonstrate strong performance in training environments but suffer degraded performance when transferred to unseen scenes, limiting their generalization capability. In contrast, VLMs leverage large-scale pre-trained vision-language knowledge to enable zero-shot or few-shot adaptation to novel scenes and objects, providing superior scene transferability and generalization[1-2].
> - Semantic understanding for complex scenarios: Current obstacle avoidance systems predominantly operate in environments with solid, opaque obstacles. However, in scenarios containing transparent objects such as glass, VLM's semantic understanding capabilities provide substantial advantages (for instance, prompts can explicitly specify that glass surfaces are obstacles requiring avoidance), whereas RL approaches face significant limitations in such cases.
>
> [1]. Yokoyama N, Ha S, Batra D, et al. Vlfm: Vision-language frontier maps for zero-shot semantic navigation[C]//2024 IEEE International Conference on Robotics and Automation (ICRA). IEEE, 2024: 42-48.
>
> [2]. Zhou G, Hong Y, Wang Z, et al. Navgpt-2: Unleashing navigational reasoning capability for large vision-language models[C]//European Conference on Computer Vision. Cham: Springer Nature Switzerland, 2024: 260-278.
>
> ---
>
> **Q2: Add the scene with dynamic obstacles**
>
> A2: Thank you for your professional and detailed comments.
>
> To address the reviewer's concern, we designed an extreme scenario containing exclusively dynamic obstacles to rigorously evaluate the depth encoder's robustness. While our original test scenarios included both static and dynamic obstacles, this new configuration provides a more challenging testbed. The evaluation environment is based on Scene 1 from the supplementary materials video, which features a complex mixture of irregular cylindrical obstacles and diverse geometric objects. To construct the extreme dynamic scenario, we converted all obstacles into continuously moving dynamic objects throughout the entire evaluation period. Visual demonstrations of this scenario are provided in the supplementary materials at **[**https://anonymous.4open.science/r/AutoFly-dynamic-3FA7**]**, accompanied by quantitative performance analysis.
>
> As shown in the table below, the depth encoder delivers a substantial performance improvement of nearly 10 percentage points in success rate (50.7% vs 40.8%) in the all-dynamic scenario. Notably, the model without the depth encoder exhibits an extremely high collision rate in this challenging setting, strongly validating the effectiveness and necessity of this module.
>
> | Method | SR | CR | PER |
> | --- | --- | --- | --- |
> | w | 50.7 | 28.2 | 73.9 |
> | w/o | 40.8 | 37.7 | 71.1 |

---

> ### Author Response · Authors · 2025-11-22
> **The Response to Reviewer nXwK (Part 2/4)**
>
> **Q3: The use of initial position**
>
> A3: Thanks for your careful review.
>
> The reviewer raises an important point. The initial position configuration is indeed critical in determining task difficulty.
>
> In our experimental setup ($70m \times 70m$), we initialize each episode by sampling a start position $s$ and a destination $d$ at the environment boundaries. This determines the initial action vector $[v_{xy}, v_z, \omega]$, where the linear velocities ($v_{xy}, v_z$) are set to zero, while the yaw rate $\omega$ is computed via proportional control to orient the UAV from $s$ toward $d$. However, due to control dynamics (particularly overshoot), this initialization yields only an approximate heading rather than perfect alignment. Following this initialization phase, the system immediately enters fully autonomous closed-loop navigation, where the AutoFly model $\pi^ * $ continuously processes visual observations $o$ and language instructions $l$ to generate flight control commands: $\pi^*: (o, l) \rightarrow a$.
>
> We deliberately designed this scenario to be challenging. By randomly sampling positions on opposite boundaries, we create straight-line traversal distances ranging from 70m to approximately 99m (diagonal side). This design ensures a rigorous evaluation of the model's autonomous navigation capabilities, requiring it to handle long-horizon flight and correct for initial heading approximations under demanding conditions.
>
> ---
>
> **Q4: Two-stage training paradigm details**
>
> A4: Thanks for your comments.
>
> We provide detailed training specifications for our two-stage training paradigm:
>
> **Stage 1 (Vision-Language Alignment):** We follow the configuration of Prismatic-VLMs' prism-dinosiglip+7b model, using its pretrained weights to initialize the MLP Projector and LLM. For comprehensive training details of this stage, we refer readers to the Prismatic-VLMs implementation.
>
> **Stage 2 (Robot Action Fine-tuning):** We freeze the Depth Generator component within the Pseudo Depth Encoder while training the following modules: Depth Projector, MLP Projector, LLM, and Visual Encoder. The hyperparameters are configured as follows:
>
> - Learning rate: 1e-4 for the Depth Projector; 2e-5 for all other trainable modules
> - Scheduler: Constant
> - Optimizer: AdamW
> - Training steps: 80,000 steps (approximately 8 epochs)
> - Batch size: 32
> - Hardware: 8 NVIDIA A100 GPUs 80G
>
> ---
>
> **Q5: The use of loss function**
>
> A5: Thanks for your comment.
>
> We sincerely apologize for the typographical error in the manuscript. Following the approach of RT-2 and OpenVLA, we employ the standard cross-entropy loss commonly used in language models, not the L1-norm loss as incorrectly stated. We have corrected this error in the revision version. The action output follows an autoregressive generation paradigm with the following implementation:
>
> Each action dimension is first normalized to the range [-1, +1] and uniformly discretized into 256 bins. These 256 bins are then mapped to the last 256 tokens in the LLaMA-2 vocabulary, establishing a discrete action-to-language mapping. This tokenization scheme enables us to leverage the standard language model training paradigm for action prediction. During inference, the predicted tokens are inversely mapped back to continuous action values.

---

> ### Author Response · Authors · 2025-11-22
> **The Response to Reviewer nXwK (Part 3/4)**
>
> **Q6: What causes the task success rate to be only 50%-60%? The experiment needs to analyse more failure cases to explain the rationality of the success metric settings and the effectiveness of the depth encoder.**
>
> A6: Thank you for your professional, detailed, and constructive review.
>
> **Analysis of factors contributing to the 50-60% success rate:** Our failure case analysis reveals that beyond collisions, the primary cause of the moderate success rate is exceeding the step limit, predominantly due to path planning failures. This challenge stems from the deliberately constrained information setting: the model receives only coarse-grained language instructions, visual observations, and an initial approximate direction (encoded in the initial action $\omega$), without any additional information such as maps, GPS coordinates, or semantic scene graphs. This minimal information paradigm significantly increases the planning difficulty, which is precisely the challenging scenario our method aims to address, enabling autonomous navigation that integrates obstacle avoidance, recognition, and planning under severely limited information.
>
> **Rationality of the success metric settings:** We have re-conducted comprehensive statistics and analysis on all experimental outcomes. The results confirm that all cases can be exhaustively categorized as either successful completions or failures belonging to two distinct types: collisions and exceeding the maximum step limit (with step-limit failures predominantly caused by path planning failures). These categories collectively account for 100% of all experimental outcomes, validating the comprehensiveness and rationality of our success criteria.
>
> **Effectiveness of the depth encoder:** We conducted a comprehensive re-analysis of failure cases, performing controlled comparisons of identical scene-target pairs with and without the depth encoder. To provide concrete evidence of the depth encoder's effectiveness, we present three representative challenging scenarios in the table below, demonstrating consistent improvements across multiple metrics:
>
> | Method | Dense cylinders scene (SR/CR/PE) | Dense forest scene (SR/CR/PER) | **Dynamic obstacle scenarios** (SR/CR/PER) |
> | --- | --- | --- | --- |
> | w | 57.2/21.1/78.3 | 53.6/23.7/77.3 | 50.7/28.2/73.9 |
> | w/o | 49.3/28.7/76.1 | 45.9/31.1/75.6 | 40.8/37.7/71.1 |
>
> Analysis of representative failure cases of our method and baseline (without depth encoder):
>
> - Dense cylinders scene: This scenario features numerous closely-spaced cylindrical obstacles of varying sizes. The baseline model without the depth encoder exhibits two primary failure patterns: (a) frequent collisions when attempting to navigate through narrow gaps (CR: 28.7%), and (b) overly conservative path planning that leads to excessive detours and step limit failures. In contrast, the depth encoder enables better spatial reasoning about obstacle distances and gaps, allowing the UAV to confidently identify safe passages. This results in a 7.9% improvement in success rate and a 7.6% reduction in collision rate.
> - Dense forest scene: This environment contains irregularly distributed obstacles with complex geometries resembling natural forest structures. The baseline model struggles in this visually complex setting, leading to hesitant navigation behavior and a high collision rate (31.1%). The depth encoder's geometric understanding significantly improves performance, reducing collisions by 7.4% and improving planning efficiency by 1.7%. The model demonstrates more decisive navigation and better trajectory optimization.
> - Dynamic obstacle scenarios: As detailed in Q2, this extreme test case with exclusively moving obstacles represents the most challenging evaluation setting. The baseline model's collision rate reaches 37.7%, with the UAV frequently failing to predict obstacle trajectories or maintain safe distances. The depth encoder enables predictive spatial reasoning, allowing the model to anticipate obstacle movements and plan avoidance maneuvers proactively. This results in the most substantial improvement: a 9.9% increase in success rate and a 9.5% reduction in collision rate, strongly validating the module's robustness in dynamic environments.
>
> Additionally, we present the side-by-side visualizations comparing successful (with depth encoder) and failed (without depth encoder) navigation attempts for these representative cases in the video **[**https://anonymous.4open.science/r/AutoFly-depth-75C7**]**. These qualitative demonstrations will provide intuitive visual evidence of how the depth encoder enhances both collision avoidance and path planning capabilities.

---

> ### Author Response · Authors · 2025-11-22
> **The Response to Reviewer nXwK (Part 4/4)**
>
> **Q7: instruction generation in the dataset**
>
> A7: Thanks for your comment.
>
> Our dataset employs coarse-grained language instructions generated through a systematic process. We provide GPT with specific scene elements and target objects as input, which then generates natural language commands. For our dataset spanning 12 scenes and 60 target objects, we generated 8 linguistically diverse paraphrases per scene-target pair to ensure vocabulary richness and variability. This process yielded a total of 5,760 unique instructions containing 21,680 words, providing sufficient linguistic diversity to train the vision-language alignment capabilities of our model.

---

> ### Comment · Reviewer_nXwK · 2025-11-24
>
> Thank the authors for the thorough rebuttal. Most of the doubts have been resolved. I would like to add two questions: 1.For Figure 9, it seems the drone's movement doesn't consider the z-axis, and it's difficult to see any comparison with the data quality of professional pilots. 2.The depth estimation has a limited range outdoors (around 10m). Is this method suitable for higher flight altitudes?

---

> > ### Author Response · Authors · 2025-11-24
> > **Thank you**
> >
> > Thank you very much for your response and for confirming that your major concerns have been addressed. We are grateful for your valuable insights throughout this process, as they have been instrumental in helping us significantly improve the manuscript, and we would be honored to earn your further support. We would greatly appreciate it if you could improve the score accordingly. Thank you again for your time and effort in reviewing our work!

---

> > ### Author Response · Authors · 2025-11-25
> > **The Response to Reviewer nXwK**
> >
> > We thank reviewer nXwK for the valuable time and constructive feedback.
> >
> > ---
> >
> > **Q1: About movement of z-axis and  data quality**
> >
> > A1: Thank you for your insightful and constructive comments.
> >
> > We respectfully note that you may be referring to Figure 10.
> >
> > **Z-axis movement:** Thank you for your insightful observation. We would like to clarify that z-axis movement is indeed considered in our approach. AutoFly's output actions include $ v_z $, and the system performs comprehensive obstacle avoidance and planning in full three-dimensional space.
> >
> > **Data quality comparison:** The trajectories generated by our RL-trained agent show minimal differences compared to those flown by professional pilots in terms of obstacle avoidance and path planning. The only difference is that the agent exhibits slight height fluctuations when navigating over larger obstacles. However, these fluctuations have no practical impact on deployment for several reasons: (1) their magnitude is well within the tolerance of modern flight control systems, (2) they do not affect the safety margins or obstacle clearance, (3) the low-level flight controller is designed to stabilize such perturbations through feedback control, and (4) professional pilots also exhibit similar vertical velocity variations when crossing large obstacles, as such dynamic adjustments are inherent to reactive obstacle avoidance. These minor variations represent a reasonable trade-off between trajectory smoothness and robust obstacle avoidance in the learned policy.
> >
> > We sincerely apologize for any confusion caused by the viewing angle of the figure. To address this concern, we have added an enhanced visualization comparison **[**https://anonymous.4open.science/r/AutoFly-more-0C7A**]** (part 2) that more clearly demonstrates the z-axis trajectory movement and data quality comparison. Additionally, we have included a video demonstration in **[**https://anonymous.4open.science/r/AutoFly-more-0C7A**]** (part 1) showing the UAV crossing obstacles, which clearly illustrates the vertical movement along the z-axis. These results further validate the reliability of our large-scale trajectory data generation approach.
> >
> > ---
> >
> > **Q2: About higher flight altitudes for AutoFly**
> >
> > A2: We thank the reviewer for their good comments.
> >
> > Our method is well-suited for higher flight altitudes. AutoFly takes both RGB images and depth maps as input and makes decisions by integrating both modalities. At higher altitudes, there are typically minimal obstacles, which means limited or no depth information is available. In such scenarios, AutoFly does not need to perform continuous obstacle avoidance and planning, and can focus primarily on locating the target described by the language command. In the rare case where obstacles do exist at high altitudes, the scenario becomes similar to our standard low-altitude settings. When obstacles are distant, AutoFly can leverage the RGB modality to perceive environmental information and make informed navigation decisions, demonstrating the robustness of our multi-modal fusion approach.
> >
> > To validate this capability, we conducted experiments at an altitude of 40m, with the target object also positioned at 40m. We placed 2 distractor objects around the target to create a more challenging scenario. The UAV performed autonomous navigation under these high-altitude conditions across 20 test trials. As shown in the table below, AutoFly achieved a high success rate even in scenarios with minimal depth information, demonstrating its strong scene generalization capability across different altitude regimes. We provide the video demo link in **[**https://anonymous.4open.science/r/AutoFly-more-0C7A**]** (part 3)
> >
> > | Method | SR | CR | PER |
> > | --- | --- | --- | --- |
> > | AutoFly | 75.0 | 0 | 88.6 |

---

### Meta-Review · Area_Chair_pAyH · 2026-01-06

**Summary:**

This paper presents AutoFly, a Vision-Language-Action (VLA) model designed for aerial vision-language navigation. To enhance 3D spatial understanding, the method incorporates pseudo-depth encoding, derived from monocular depth estimation, into the OpenVLA architecture. Furthermore, the authors construct a novel dataset using the AirSim simulator, employing reinforcement learning to generate expert demonstration trajectories that emphasize continuous obstacle avoidance and autonomous planning. The discussion centered on the depth encoder’s efficacy and the model’s robustness in challenging real-world scenarios, such as dynamic outdoor environments. Reviewers also emphasized the importance of benchmarking against recent baselines like SPF and validating the reliability of sim-to-real performance.

**Reviewer Concerns:**

The authors further clarified the ablation study of the depth encoder and provided additional details on the training paradigm. They also conducted new experiments in more challenging scenarios, completing additional comparisons and validation of the method. All reviewers indicated that their concerns were satisfactorily addressed.

**Reviewer Scores:**

Following the rebuttal, the consensus improved, with one reviewer raising their score to 8 after the authors clarified the concerns raised. Overall, the paper provides a substantial contribution to UAV vision-language navigation research, given the strong majority support and the effective resolution of key issues, the Area Chair recommends accepting the paper.

---

### Decision · Program_Chairs · 2026-01-26

Accept (Poster)